# Moderate global warming does not rule out extreme global climate outcomes

Emanuele Bevacqua[1✉], Erich Fischer[2], Jana Sillmann[3,4] & Jakob Zscheischler[1,5]

Effectively communicating worst-case projections of global future climate—hereinafter referred to as worst-case climate outcomes—is essential for risk assessment and developing robust adaptation strategies to global warming[1–7]. Yet, current approaches for identifying spatially consistent climate outcomes are limited, with worst-case global climates typically communicated via the average of climate model projections at high global warming levels, such as 3 °C or 4 °C above the preindustrial era[8,9]. Here we show that extreme global climate outcomes may occur even under moderate 2 °C warming for several sectors. For droughts in global key breadbasket regions, precipitation extremes over highly populated areas and fire weather extremes across forests, global climatic impact-drivers at 2 °C of global warming may turn out to be much more extreme than model-averaged projections at 3 °C or 4 °C warming. We derive these results by identifying sector-specific, spatially consistent potential high- and low-impact global climate outcomes through spatially averaging projected sector-relevant climatic impact-drivers across key global regions. Our approach can easily be adapted to a wide range of sectors to support the improvement of sector-specific climate risk assessment and to inform climate policy. As global warming approaches 1.5 °C (ref. 10), these findings underscore the urgency of rapid mitigation to limit warming well below 2 °C, as even a 2 °C world may entail severe impacts.

Robust risk management under high uncertainty requires careful consideration of extreme outcomes, which can result in strong socioeconomic and/or environmental repercussions[1–4]. In the context of climate change, large uncertainties in projections[11] highlight the potential for high-impact global future climate outcomes[2,6]. Such large uncertainties warrant a departure from solely focusing on the most likely climate outcomes[5,7]—usually represented by the multimodel mean of climate model simulations—as this approach can leave socioeconomic and environmental sectors either very vulnerable or excessively prepared to inconceivable yet plausible extreme global climate outcomes[6,7,12,13].

Yet, our understanding of potential extreme global-scale climate remains limited[2,14], as underscored by the Safe Landing Climates Lighthouse Activity of the World Climate Research Programme, which calls for new methodologies to "identify risks from low-probability, high-impact possibilities with global-scale ramifications". So far, worst-case global climate outcomes are typically explored—for instance, by the Intergovernmental Panel on Climate Change (IPCC)—by studying averages of multiple climate model simulations reaching a high global warming level, such as 3 °C or 4 °C (ref. 8). Alternatively, climate models with high climate sensitivity—that is, models that reach high warming levels—are used to explore end-of-century extreme conditions[9]. However, uncertainties in projections of climatic impact-drivers exist at any warming level[15] owing to climate model differences and internal, natural climate variability[7,13]. Consequently, focusing on extreme global warming levels to explore and communicate high-impact global climate risks may hide the potential for extreme climate outcomes for specific sectors at much more moderate warming levels, such as +2 °C. Global maps at a given warming level that display the most extreme projected outcome at each location in variables such as temperature, precipitation and soil moisture[16] can provide information on worst-case outcomes at the local scale[15]. However, these maps do not represent plausible global patterns[16] as they do not take spatial dependencies into account and are thus not spatially coherent.

Here we present an approach to assess whether global climate-sensitive sectors may face potential high-impact climate outcomes even under moderate warming levels. This is important for accurately quantifying and communicating risks in the context of the Paris Agreement's warming targets, where moderate global warming might give a false sense of security. We illustrate the approach for five different global sectors and then discuss future perspectives for implementing the approach in other sectors and integrating it into risk assessments.

## Identifying extreme climate outcomes

We derive global climate outcomes from publicly available climate model simulations of the Coupled Model Intercomparison Project Phase 6 (CMIP6)—the same models routinely used in IPCC reports to inform policymakers[6,8,12,17,18] (Methods). Given a set of available future

[1]Department of Compound Environmental Risks, Helmholtz Centre for Environmental Research - UFZ, Leipzig, Germany. [2]Institute for Atmospheric and Climate Science, Department of Environmental Systems Science, ETH Zurich, Zurich, Switzerland. [3]Research Unit for Sustainability and Climate Risks, University of Hamburg, Hamburg, Germany. [4]CICERO Center for International Climate Research, Oslo, Norway. [5]Department of Hydro Sciences, TUD Dresden University of Technology, Dresden, Germany. ✉e-mail: emanuele.bevacqua@ufz.de

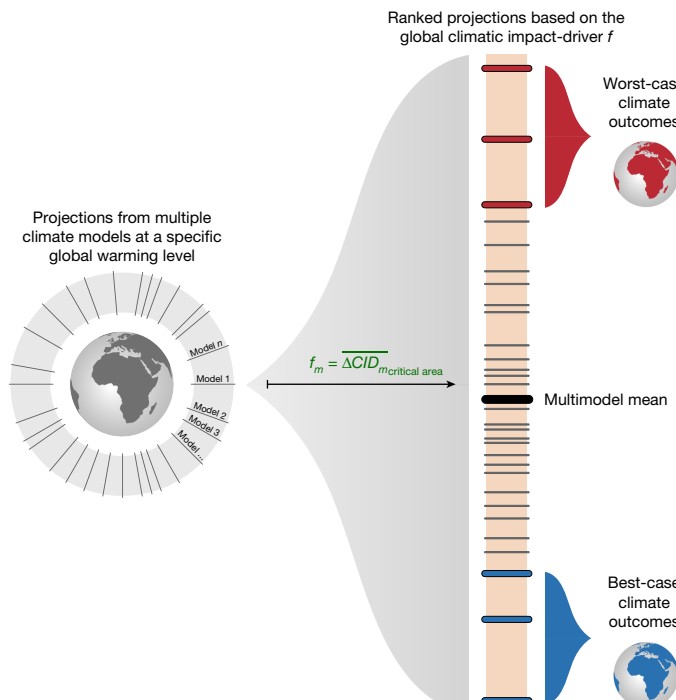

**Fig. 1 | Approach for identifying extreme future global climate outcomes from available climate model simulations.** For a given sector, projections from multiple climate models at a specific global warming level (in this paper, 2 °C) relative to preindustrial climate conditions (on the left) are ranked (on the right) using the global sector-mean projected climatic impact-driver (abbreviated as global climatic impact-driver), which serves as a proxy for projected potential impacts on the sector. For each climate model $m$, the global climatic impact-driver averages a projected sector-critical climatic impact-driver (relative to preindustrial conditions; $\Delta CID_m$) across the critical area over which the sector operates. For example, for the global food security sector, $f$ may quantify the projected drought frequency across key breadbasket regions relative to preindustrial conditions. The ranked model projections define climate outcomes, ranging from those with the highest to lowest values of $f$, referred to as worst- and best-case climate outcomes.

climate model simulations at a specific level of global warming, we define extreme climate outcomes relative to a specific sector (Fig. 1). Different sectors are sensitive to different climatic impact-drivers[19], and what constitutes an extreme climate outcome is therefore strongly sector dependent. For instance, a very dry climate can be detrimental to crop production but less impactful to flood-risk management. Drawing from expert judgement, stakeholders and existing literature, our bottom-up approach first identifies a sector-specific local climatic impact-driver and a critical area over which the sector operates. For example, droughts in key breadbasket regions could reduce crop yields and thus threaten global crop production and food security. This information is then used to construct the global sector-mean projected climatic impact-driver, $f$ (hereinafter, global climatic impact-driver; Fig. 1). Specifically, the global climatic impact-driver is computed for each climate model $m$ as the spatially weighted average of the projected local climatic impact-driver relative to preindustrial climate conditions ($\Delta CID_m$) across the critical area, where each grid cell $x$ is weighted by its area $a_x$:

$$f_m = \overline{\Delta CID}_{m\,\text{critical area}} = \frac{\sum_{x \in \text{critical area}} a_x \, \Delta CID_{x,m}}{\sum_{x \in \text{critical area}} a_x} \quad (1)$$

As the global climatic impact-driver $f$ is a scalar, which serves as a proxy for projected potential impacts on the specific sector, we can rank the climate model simulations accordingly. We define extreme climate

outcomes as simulations with the upper and lower approximately 10% values of $f$ (8% or 12%, depending on the model availability for the sector; Methods). To distinguish between high- and low-extreme climate outcomes, we refer to them as worst-case and best-case models or climate outcomes. However, as 'worst' and 'best' imply a value judgement, they should be interpreted cautiously, particularly because these classifications are sector specific. This is in line with the IPCC definition of climate risk, referring to risk as "the potential for adverse consequences for human or ecological systems, recognizing the diversity of values and objectives associated with such systems"[8]. Moreover, the global climatic impact-driver identifies climate outcomes that are extreme on average across a large critical area, meaning that a worst-case climate outcome may still be moderate in some sub-areas. Finally, $f$ quantifies sector-relevant climatic impact-drivers across critically exposed areas, whereas impacts and associated risks arise from the interaction of climatic impact-drivers with exposure and vulnerability.

Here we use the approach to identify extreme climate outcomes for: (1) precipitation extremes in highly populated areas, which may induce devastating floods; (2) concurrent droughts in global breadbaskets, which threaten food security; and (3) fire weather extremes across the world's forests, which put critical ecosystem services at peril. Two additional sectors will be briefly discussed in the final section.

## Flood-inducing rain in populated areas

Pluvial floods in urban areas—surface-water flooding from intense rainfall that exceeds natural or engineered drainage capacity—are responsible for a substantial share of the total impacts from coastal and inland flooding, ranking among the most impactful natural hazards[20]. Because pluvial floods are typically triggered by precipitation extremes accumulating over one or a few days, such extremes are a key proxy for flooding in climate studies[21]. Owing to the higher water-holding capacity of a warmer atmosphere[11], maximum consecutive 5-day precipitation (Rx5day) is projected to intensify across nearly all landmasses in a 2 °C warmer world relative to preindustrial conditions[21,22], as indicated by the multimodel mean of climate simulations[22] (Fig. 2a). However, projections of precipitation extremes are uncertain owing to uncertain changes in moisture availability and, particularly, atmospheric circulation changes[11]. Thus, the multimodel mean of precipitation extremes across the globe, although widely used to illustrate projections of extremes with increasing global warming, may hide much more moderate or extreme projections of precipitation extremes[11,12]. To inform on impact-relevant events, here we consider only precipitation extremes across populated areas, as human settlements are a prerequisite for most flooding impacts. Such exposed populated areas—where vulnerability to flooding depends on factors such as drainage capacity—are also relevant for the reinsurance sector, which typically focuses on exposed areas to assess financial risks[20,23].

To isolate global climate outcomes associated with projected precipitation extremes across populated areas, we use a global climatic impact-driver $f$ that, for each climate model simulation $m$, averages the projected 5-day precipitation annual maxima relative to preindustrial conditions ($\Delta Rx5day$ (%)) over grid cells that are in highly populated areas (Methods), that is $f_m = \overline{\Delta Rx5day_m(\%)}_{\text{populated area}}$. In a 2 °C world, heavy precipitation projections over populated areas vary widely across climate models, ranging from an increase of 4% to 15% (left orange bar in Fig. 2b). Worst- and best-case outcomes, represented by models with the highest and lowest values of the global climatic impact-driver $f$, respectively, are highlighted with thick coloured lines (see Extended Data Table 1 for values of all models). The large uncertainty in $f$ is reflected in substantial differences between the spatially explicit averages of worst-case (Fig. 2c) and best-case (Fig. 2d) outcomes. It is noted that although precipitation extremes intensify virtually everywhere in the worst-case outcomes (Extended Data Fig. 1g–i), spatial differences exist between the individual worst-case (or best-case) outcomes

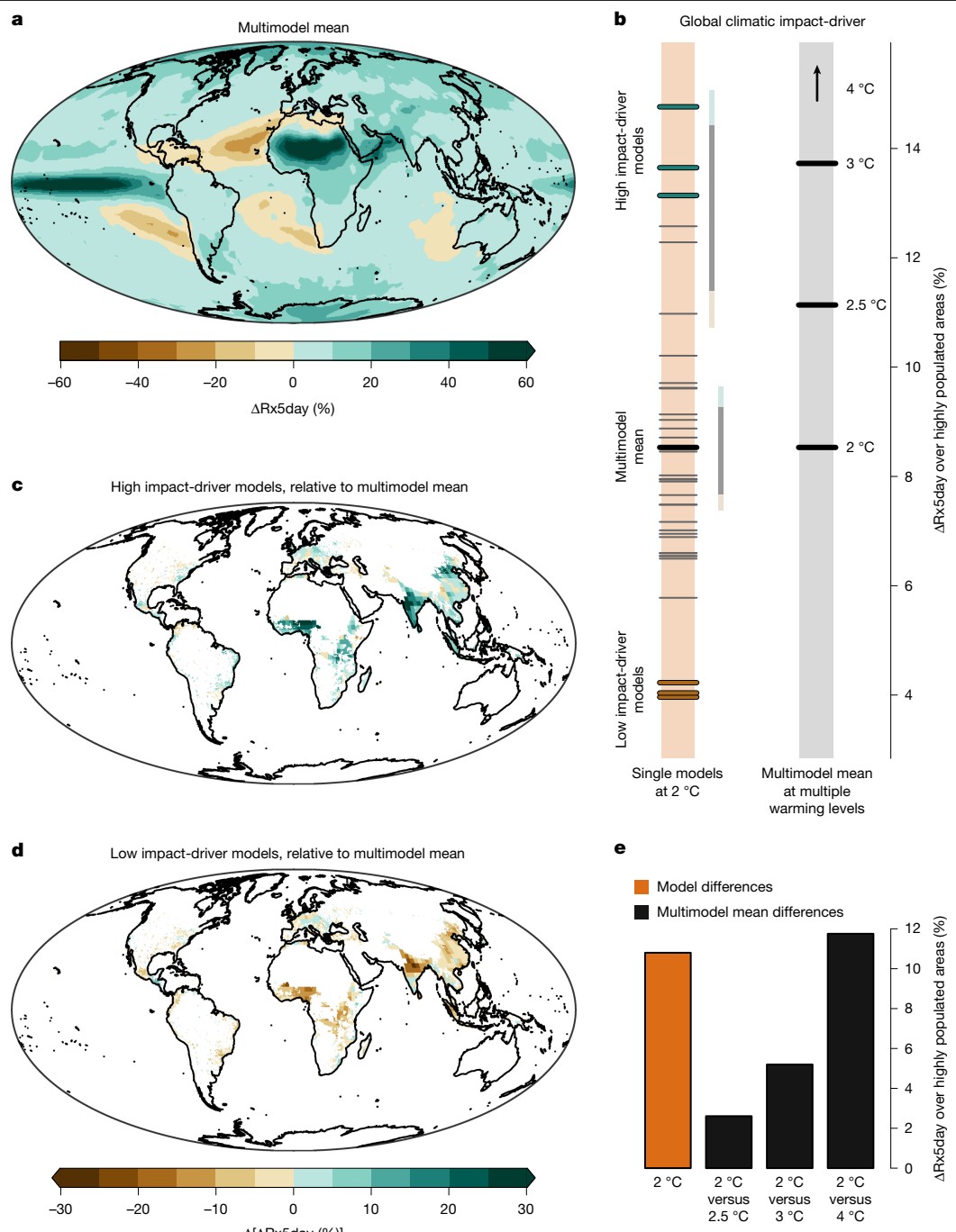

**Fig. 2 | Extreme climate outcomes for precipitation extremes in highly populated areas. a**, Multimodel mean of the projected mean annual maximum consecutive 5-day precipitation in a 2 °C warmer world relative to preindustrial conditions (1851–1900), that is, ΔRx5day (%). **b**, Left orange bar: the global climatic impact-driver *f* used to identify climate outcomes, that is, ΔRx5day (%) averaged over highly populated areas (areas shown in **c** and **d**), computed for individual climate models in a 2 °C world (the black thick line shows the multimodel mean). The models with the highest and lowest 8% values of *f* represent the worst and best cases (from top to bottom: CanESM5, KACE-1-0-G and IPSL-CM6A-LR (worst) and GFDL-ESM4, MRI-ESM2-0 and INM-CM4-8 (best); see Extended Data Table 1 and Extended Data Fig. 2a for model names). Note that one model (horizontal grey line) lies almost entirely beneath the

lowest thick worst-case line due to very similar *f* values. The two vertical lines to the right of the bar show the range of *f* for two SMILEs (coloured for the highest and lowest 8% values—see Extended Data Fig. 2a for more SMILEs). Right bar: multimodel mean of *f* at 2 °C, 2.5 °C, 3 °C and 4 °C global warming levels (the arrow indicates values above the *y*-axis range). **c**, Worst-case climate outcome at 2 °C, that is, the difference between ΔRx5day (%) averaged across worst-case models and the multimodel mean, displayed over highly populated areas. **d**, The same as in **c** but for the best-case outcome. **e**, Range of *f* values across model simulations in a 2 °C world (left orange bar); and difference between multimodel mean of *f* at 2 °C and 2.5 °C (second bar), 2 °C and 3 °C (third bar), and 2 °C and 4 °C (fourth bar). Data from: climate models, ref. 51; population, ref. 52.

(Extended Data Fig. 1). Our results also indicate that model improvement could reduce uncertainties in the global climatic impact-driver, as they are largely driven by structural differences between models rather than by internal climate variability (Fig. 2b, the total uncertainty shown by

the model range on the left orange bar is large compared with the uncertainty from internal climate variability shown by the range of each of the two Single Model Initial-condition Large Ensembles (SMILEs); see Extended Data Fig. 2a and Extended Data Table 2 for more SMILEs).

The worst-case climate outcome is particularly extreme, with precipitation extremes across populated areas at a moderate 2 °C warming projected to exceed the multimodel mean at 3 °C of global warming (Fig. 2b). The potential for such a worst-case climate outcome in a 2 °C world is consistent with the uncertainty in projections of precipitation extremes at 2 °C of global warming, which is nearly as large as the difference between the multimodel mean response at 2 °C and 4 °C (Fig. 2e).

## Global breadbasket failures from drought

Very low global crop production owing to concurrent crop failures in major breadbaskets worldwide can impact global food security and supply chains, especially as international trade of consumable food rises and an increasing number of people rely on imported food[24,25]. Concurrent droughts across breadbaskets are often used as an indicator to assess future climate risks to global agriculture and food security[26]. The frequency of annual soil moisture droughts is projected to increase in most regions according to the multimodel mean, with the main exception of central Africa and east of the Caspian Sea (Fig. 3a).

To isolate global climate outcomes associated with projected droughts in crop-producing regions, we use a global climatic impact-driver $f$ that, for each climate model simulation $m$, averages the projected drought frequency relative to preindustrial conditions ($\Delta$Drought frequency) across global breadbaskets, that is, $f_m = \overline{\Delta\text{Drought frequency}}_{m_{\text{breadbaskets}}}$. The considered breadbaskets[25] cover most of the global maize, wheat, soybean and rice productions. In a 2 °C world, the average drought frequency across breadbaskets may increase by more than 50%—meaning a shift from 20% in the preindustrial period to more than 70% in the future—or even remain unchanged (left orange bar in Fig. 3b; Extended Data Table 1). This very large uncertainty in the global climatic impact-driver $f$ mainly arises from model differences rather than internal climate variability (Fig. 3b, compare the range from models over the left orange bar with the range from two SMILEs; see Extended Data Fig. 2b and Extended Data Table 2 for more SMILEs). The large inter-model differences in $f$ are reflected by a large difference in the spatially explicit average of worst-case models (Fig. 3c) and best-case models (Fig. 3d; see Extended Data Fig. 3 for individual models), with the former far above the multimodel mean at a 2 °C warming in almost all breadbasket regions.

Out of 42 models, 10 models show climate outcomes at a 2 °C warming that are well beyond the multimodel mean at 4 °C of global warming (Fig. 3b). This stresses the need for a careful assessment of their plausibility by evaluating the underlying physical processes. Yet, regardless of pending plausibility checks, these results also call for examining the potential impacts of extreme climate outcomes on global crop production through crop modelling, which can incorporate factors such as irrigation, soil properties and complex management practices. At the same time, it should also be noted that many model simulations indicate only very small changes in drought frequency. Consistent with the above, the differences between the multimodel mean response at different global warming levels are negligible compared with the uncertainty from model differences at 2 °C warming (Fig. 3e). These results suggest that even in the case of potentially reaching confidence about future global warming—by narrowing climate-sensitivity estimates or adopting decisive climate policies—uncertainties in future food security will remain large.

## Global forest wildfires from fire weather

Weather conditions conducive to wildfires are projected to intensify across most landmasses in a 2 °C warmer world (Fig. 4a). Such a broad intensification of wildfire risk is particularly worrying for forested regions[27], which cover nearly a third of the global land area and serve as crucial carbon sinks by absorbing atmospheric carbon dioxide through photosynthesis, thereby slowing anthropogenic climate change.

Increased wildfire activity leads to extensive burned areas, carbon loss and heightened emissions, with far-reaching consequences worldwide[27]. Although human activities influence fire dynamics[28], fire-prone weather conditions are a key driver of wildfires. Over the past two decades, global carbon emissions associated with forest fires have increased by 60%, primarily driven by more frequent fire-favourable weather and increased vegetation productivity in extratropical regions[27]. At the same time, anthropogenic climate change has already increased the global burned area by about 16% (ref. 29). Given these trends, it is important to understand possible ranges of fire-conducive weather conditions across forests in the near future.

We use a global climatic impact-driver $f$ that, for each climate model simulation $m$, averages the projected daily fire weather index (FWI) annual maxima[30] relative to preindustrial conditions ($\Delta$FWIx) over forests, that is $f_m = \overline{\Delta\text{FWIx}}_{m_{\text{forests}}}$. Climate models indicate a wide range of projections in FWI over forests in a 2 °C world, with the worst-case model showing an increase more than 4-times larger than the best-case model (+6.5 against +1.5 relative to preindustrial conditions, left orange bar in Fig. 4b; Extended Data Table 1). This large difference in $f$ across climate model simulations is largely caused by model differences rather than internal climate variability (Fig. 4b; Extended Data Fig. 2c and Extended Data Table 2 for more SMILEs), which indicates the potential for model improvements and uncertainty reduction. As a result, the averages of the best- and worst-case climate outcomes largely deviate from each other (Fig. 4c,d and Extended Data Fig. 4).

The four worst-case models at a moderate 2 °C warming show an increase in FWI extremes across forests larger than the multimodel mean projection at 3 °C warming (Fig. 4b)—this occurs despite, owing to limited data availability, about half as many models being used for the forest wildfires sector compared with the other sectors. Accordingly, the difference between the multimodel mean response of FWI extreme at different warming levels is similar or even smaller compared with the uncertainty from model differences at 2 °C warming (Fig. 4e). Therefore, as for the other sectors discussed above, for FWI extremes across forests, considering uncertainties in projections at 2 °C of global warming is essential for robust risk assessments under moderate warming levels.

## Factoring worst cases into decisions

A common perception is that worst-case future global climates are associated with extremely high global warming levels[9]. However, we show that for frequently studied climatic impact-drivers across critical globally relevant sectors, this perception is not accurate. Large uncertainties in climate model projections, mostly stemming from model differences, are evident even at moderate future warming and can lead to sector-specific high-impact global climate outcomes. These findings reinforce previous research emphasizing the need to limit global warming well below 2 °C to avoid extreme climate outcomes[31].

Although it is known that climate projections for specific locations have large uncertainties at a given global warming level[15], our findings shed light on how these local uncertainties propagate into globally averaged projections and allow for extreme global climate outcomes. Owing to large uncertainties at the local scale, projected local climatic impact-drivers at 2 °C warming can largely exceed the multimodel mean projection at 3 °C or even 4 °C of global warming (Extended Data Fig. 5). However, this large local uncertainty does not directly inform about global-scale worst-case climate outcomes, as uncertainty is strongly reduced by global-scale averages. Accordingly, the unrealistic assumption that every location may experience the worst climate outcomes simulated locally by different models strongly overestimates the global-scale worst case. Similarly, this approach would result in a too optimistic global-scale best case. Specifically, constructing unrealistic global worst- and best-case climate outcomes at 2 °C of warming in this way (Extended Data Fig. 6 and Methods) inflates uncertainty

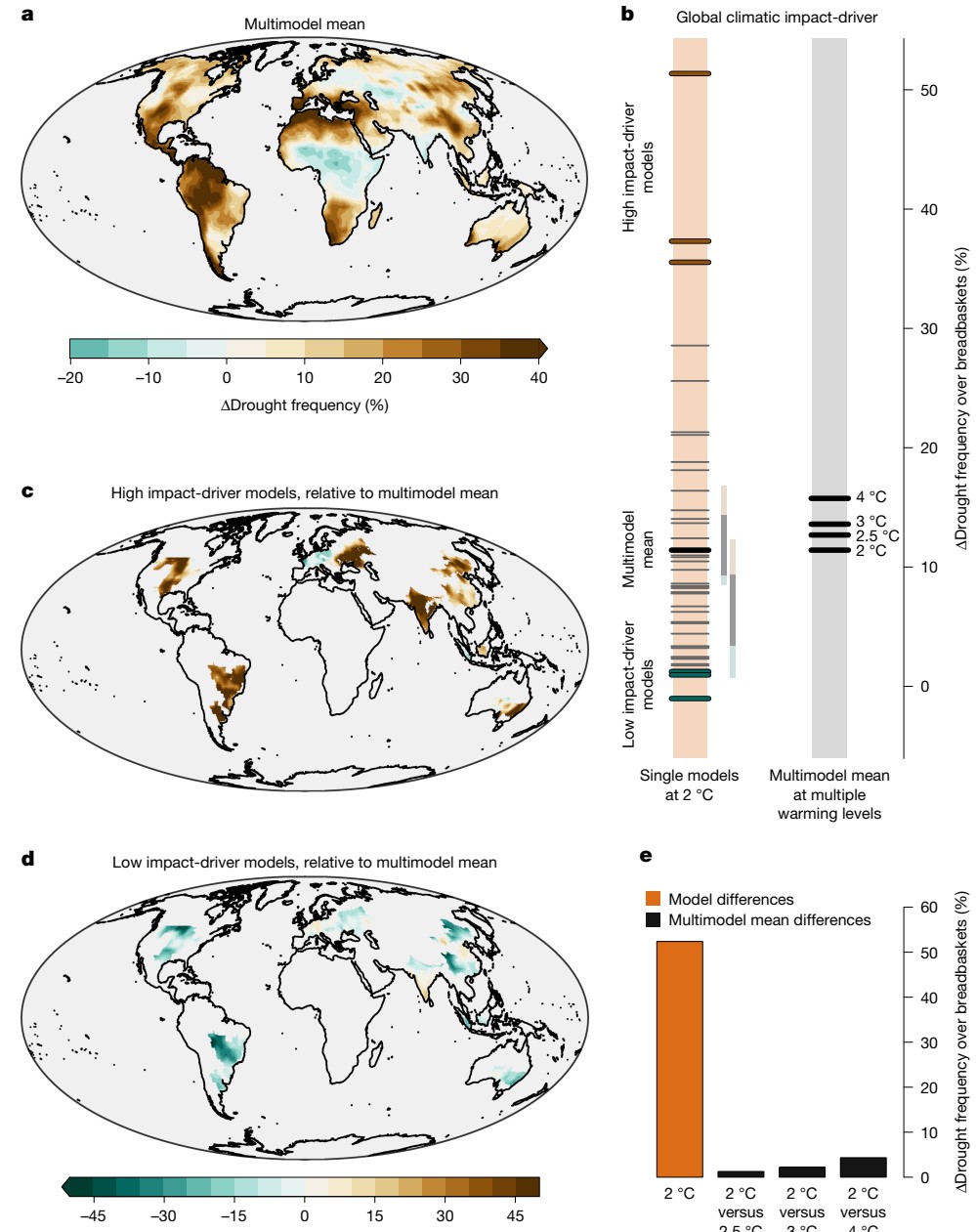

**Fig. 3 | Extreme climate outcomes for droughts across breadbaskets worldwide. a**, Multimodel mean of the projected frequency of annual soil moisture droughts in a 2 °C warmer world relative to preindustrial conditions (1851–1900), that is, ΔDrought frequency. **b**, Left orange bar: the global climatic impact-driver $f$ used to identify climate outcomes, that is, ΔDrought frequency averaged over global breadbasket regions (areas shown in **c** and **d**), computed for individual climate models in a 2 °C world (the black thick line shows the multimodel mean). The models with the highest and lowest 8% values of $f$ represent the worst and best cases (from top to bottom: CNRM-ESM2-1, BCC-CSM2-MR and GFDL-CM4 (worst) and FGOALS-f3-L, MCM-UA-1-0 and TaiESM1 (best); see Extended Data Table 1 and Extended Data Fig. 2b for model names). Note that one model (horizontal grey line) lies almost entirely beneath the

highest thick best-case line due to very similar $f$ values. The two vertical lines to the right of the bar show the range of $f$ for two SMILEs (coloured for the highest and lowest 8% values—see Extended Data Fig. 2b for more SMILEs). Right bar: multimodel mean of $f$ at 2 °C, 2.5 °C, 3 °C and 4 °C global warming levels. **c**, Worst-case climate outcome at 2 °C, that is, the difference between ΔDrought frequency averaged across worst-case models and the multimodel mean, displayed over global breadbasket regions. **d**, The same as in **c** but for the best-case outcome. **e**, Range of $f$ values across model simulations in a 2 °C world (left orange bar); and difference between multimodel mean of $f$ at 2 °C and 2.5 °C (second bar), 2 °C and 3 °C (third bar), and 2 °C and 4 °C (fourth bar). Data from: climate models, ref. 51; breadbaskets, ref. 25.

in the considered global projected climatic impact-driver $f$—that is, the difference between $f$ values computed on such unrealistic global worst- and best-case climate outcomes—by about 80% to 290% across the 3 sectors (Extended Data Fig. 6c,f,i). Nevertheless, we find that the actual uncertainty in $f$, properly estimated based on globally averaged projections, is still large—about the same as or even 12-times larger than, depending on the sector, the difference between the multimodel mean

at 4 °C and 2 °C of global warming (Figs. 2e, 3e and 4e). Consequently, considering model uncertainty at 2 °C warming reveals extreme global climate outcomes across critical areas that exceed the multimodel mean at 3 °C or 4 °C of global warming (Figs. 2–4)—levels of warming that are well beyond the Paris Agreement limits.

Our approach can easily be applied to other globally exposed sectors, for instance, hydro-power generation shortfalls owing to low

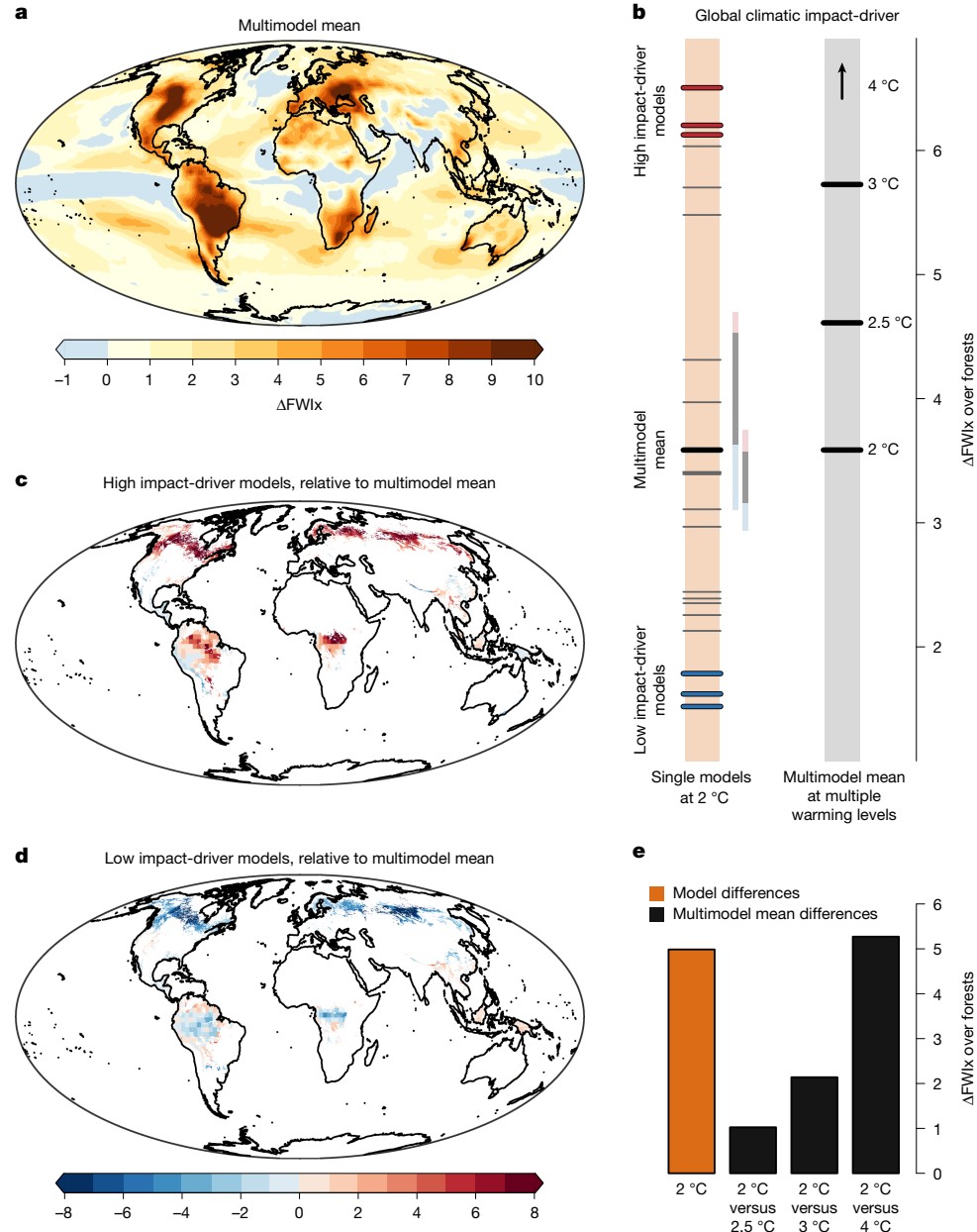

**Fig. 4 | Extreme climate outcomes for fire weather extremes across global forests. a**, Multimodel mean of the projected mean annual maximum daily fire weather index FWI in a 2 °C warmer world relative to preindustrial conditions (1851–1900), that is, ΔFWIx. **b**, Left orange bar: the global climatic impact-driver *f* used to identify climate outcomes, that is, ΔFWIx averaged over global forests (areas shown in **c** and **d**), computed for individual climate models in a 2 °C world (the black thick line shows the multimodel mean). The models with the highest and lowest 12% values of *f* represent the worst and best cases (from top to bottom: HadGEM3-GC31-MM, KACE-1-0-G and UKESM1-0-LL (worst) and EC-Earth3, MRI-ESM2-0 and KIOST-ESM (best); see Extended Data Table 1 and Extended Data Fig. 2c for model names). The two vertical lines to the right of the bar show the range of *f* for two SMILEs (coloured for the highest and lowest 12% values—see Extended Data Fig. 2c for more SMILEs). Right bar: multimodel mean of *f* at 2 °C, 2.5 °C, 3 °C and 4 °C global warming levels (the arrow indicates values above the *y*-axis range). **c**, Worst-case climate outcome at 2 °C, that is, the difference between ΔFWIx averaged across worst-case models and the multimodel mean, displayed over global forests. **d**, The same as in **c** but for the best-case outcome. **e**, Range of *f* values across model simulations in a 2 °C world (left orange bar); and difference between multimodel mean of *f* at 2 °C and 2.5 °C (second bar), 2 °C and 3 °C (third bar), and 2 °C and 4 °C (fourth bar). Data from: FWI, ref. 53; forests, refs. 54,55.

water availability, drought-induced tree mortality and coastal impacts from sea-level rise. Focusing on critical sectoral areas is essential, as averaging climatic impact-driver changes globally can mask extremes concentrated in these critical areas, leading to an underestimation of worst-case climate outcomes. Our results suggest that the more complex the climatic impact-driver affecting a sector, the larger the uncertainty and the potential for extreme climate outcomes at a given global warming level. For instance, the drought evolution depends on interacting effects, including changes in atmospheric circulation and

clouds that modulate precipitation, surface radiation and wind—factors that, together with vegetation response to increased carbon dioxide, alter evapotranspiration[9]. In contrast, sectors heavily dependent solely on temperature tend to show lower uncertainty, as temperature-related variables have less room to vary across models at a fixed global warming level. This is illustrated by projections of heatwaves in low-income countries[32], which include vulnerable regions with limited adaptation potential, such as heat warning systems, well-insulated or air-conditioned buildings, and well-equipped hospitals. Across these countries, the

second and third worst-case outcomes at 2 °C align with the multi-model mean at 2.5 °C, although the worst-case model reaches the multimodel mean at 3 °C (Extended Data Fig. 7). Projections of extreme (1-in-10 years) annual sea surface temperature averaged across global fisheries—relevant for fish mortality, fish catches, stocks and communities reliant on these resources[33,34]—appear narrow at 2 °C of global warming, ranging from 1.50 °C to 1.78 °C relative to preindustrial conditions (Extended Data Fig. 8). Yet, despite such small uncertainty at the global scale for fisheries, the averages of the best- and worst-case climate outcomes show very different spatial patterns (Extended Data Fig. 8c,d), highlighting large regional uncertainty. Moreover, our annual temperature-based impact-driver $f$ may underestimate uncertainty, as fish mortality is probably also influenced by short-term marine heatwaves, deoxygenation, nutrient shortage, pollution, overfishing and fish migrations[35–37]. This underscores the need to move beyond simple temperature-based climatic impact-drivers to capture compounding climatic drivers[38].

Extreme climate outcomes should not be disregarded by considering only the multimodel mean or narrow uncertainty ranges of climate projections. The plausibility of extreme climate outcomes should be scrutinized via process-based evaluation[39], and could include process-oriented emergent constraints[40] tailored to global climatic impact-drivers. Pending these checks, however, the potential implications of extreme climate outcomes must be acknowledged transparently and factored into decision-making. In this context, a striking result is the very large difference in global climate outcomes for droughts across global breadbasket regions. Although this finding is consistent with known local uncertainties in projections of precipitation and soil moisture dynamics[16], it emphasizes the need to improve models. At the same time, it demonstrates the need to develop strategies for dealing with potential worst cases. The general importance of considering worst-case outcomes for stress-testing adaptation measures[1–4,41] is further strengthened by ample evidence that observed trends in some key-climatic variables fall at the limit or even outside of model projections[39,42–45]. Accordingly, CMIP models constitute an ensemble of opportunity that may not accurately cover the full range of uncertainty in projections and may not sample the actual worst cases—for example, adding a few independent models could reveal more extreme climate outcomes at +2 °C. Moreover, climate models may have systematic biases, leaving room for a reality that could be more or less severe than any current projection[46]. For certain aspects, there is the hope that higher-resolution global models could eventually help to narrow down uncertainties in global climatic impact-drivers. For instance, convection-permitting global models may improve the simulation of precipitation extremes, although resolution-dependent differences in simulations are clearer in sub-daily precipitation than in the 5-day precipitation extremes analysed here[47]. More broadly, although model differences largely contribute to uncertainties, internal climate variability also has a role. Accordingly, explicitly sampling this variability reveals that climate outcomes could turn out more extreme than those captured by single-member models—both for worst cases (Extended Data Fig. 2a,d,e) and best cases (Extended Data Fig. 2d). On a positive note, simultaneous worst cases across the five considered sectors are unlikely (Extended Data Fig. 9), although this pattern is probably specific to our sectors, which rely on distinct climatic impact-drivers and critical regions. For instance, should a worst-case dry outcome occur, multiple drought-sensitive sectors would likely be affected simultaneously.

Our simple approach could be the basis for a more systematic strategy to guide impact modellers in the selection of a subset of climate models that sample the full range of possible climate outcomes[17,18,48] for a given impact sector. Currently, large-scale initiatives such as the latest protocol of the Inter-Sectoral Impact Model Intercomparison Project (ISIMIP) rely on a limited subset of climate models that likely omits the best- and worst-case climate models. Hence, ISIMIP-based simulations probably underestimate the range of possible global climate impacts for many sectors at 2 °C warming (Extended Data Fig. 10). This limitation highlights challenges in large-scale impact-modelling initiatives, possibly stemming from the high computational cost of impact simulations and the need for automated workflows that can handle large ensembles of climate model inputs. Given the need to carefully assess the consequences of worst-case climate outcomes, a concerted effort among stakeholders, impact modellers and climate scientists is essential to identify plausible evolutions of extreme global impacts that can support well-informed decision-making[49]. Our approach could also help guide a more targeted use of climate models in impact modelling and risk assessments. For example, rather than relying on uniform subsets of climate models across sectors, flood-risk or agricultural-impact modellers could—in collaboration with climate scientists and stakeholders—use our approach to identify sector-specific extreme outcomes that merit explicit simulation and to focus on global critical exposed areas. This would enable exploring currently underrepresented extreme outcomes—those that are very relevant for risk management and adaptation planning. While our indicator $f$ represents an initial step for identifying extreme climate outcomes, more advanced low-dimensional indicators—potentially derived via explainable machine learning that links climate drivers to impact data[50]—could further improve the identification of worst-case climate outcomes and optimize model selection for impact simulations. At the IPCC level, strengthening interactions between working groups I and II[1,19], which respectively assess physical climate changes and their impacts, is key for informing climate risk management and both mitigation and adaptation policy under these large uncertainties.

We have demonstrated that, across multiple sectors, global climate outcomes at 2 °C warming could be much more extreme than those expected in a 3 °C or 4 °C warmer world. As global warming approaches 1.5 °C (ref. 10), this study highlights the urgency of ambitious mitigation strategies aimed at limiting global warming well below 2 °C, emphasizing that even a 2 °C warmer world does not necessarily safeguard against severe sectoral global climate impacts.

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

## Methods

### Data

We used climate data from the Coupled Model Intercomparison Project Phase 6 (CMIP6)[51] and combined the historical scenario (1851–2014) with the Shared Socioeconomic Pathway SSP5-8.5 from 2015 onwards (Extended Data Table 1). Using SSP5-8.5 allows for inspecting warming levels from moderate (2 °C) to extreme (4 °C) with the same consistent dataset. We also used the SSP2-4.5 to test the sensitivity of the results at 2 °C of global warming and found that—except for fisheries—results derived from different SSPs are similar (Extended Data Fig. 11).

We derived time series of annual maximum 5-day precipitation (from daily data; CMIP6 identifier of the used variable: day, pr), annual mean soil moisture over the total column (from monthly data; Lmon, mrso), annual maximum of daily maximum temperature (from daily data; day, tasmax), annual mean of sea surface temperature (from monthly data; Omon, tos), and—for identifying global warming levels—global-mean temperature (from monthly data; Amon, tas). Successively, gridded time series were interpolated to an equal 2.5° spatial grid before the next analyses with bilinear remapping for tasmax and tos (for tos, conservative remapping is used for the model AWI-CM-1-1-MR owing to an original unstructured grid type), and conservative remapping for mrso and pr (no interpolation is applied to tas). For the analysis of the FWI[56,57], we used the already available annual maxima of daily FWI at 2.5° spatial grid from ref. 53, computed based on daily precipitation, wind (day, sfcWind), relative humidity (day, hurs), and maximum temperature. For all analyses, fields were interpolated to a finer grid at the end of the analyses for graphical purposes. We used one ensemble member per model: the r1i1p1f1 member when available for both the historical and the considered SSP scenario; otherwise the first ripf member in alphabetical order that was available for both historical and SSP (the ripf is an index used in CMIP6 to uniquely identify ensemble members of a given model, where r, i, p and f denote the realization, initialization method, physics parameterization and forcing index, respectively). Only when quantifying the contribution of internal climate variability to the uncertainties in projections, we used SMILEs[58,59] (Extended Data Table 2), that is, the numbers reported in the text are unaffected by the SMILEs. We used CanESM5 and MIROC6 (each with 50 ensemble members available for all analysed sectors), which are shown as vertical lines on the left and right, respectively, in Figs. 2b, 3b and 4b and Extended Data Figs. 7b and 8b; these and additional SMILEs with fewer ensemble members are shown in Extended Data Fig. 2. For each sectoral analysis individually, we selected CMIP6 model ensemble members with data starting at least in 1851 both for the specific sectoral variable and the global-mean temperature required to identify global warming levels. The resulting set of models is shown in Extended Data Tables 1 and 2. We analysed individual ensemble members of SMILEs independently, which allows for consistent comparison with results from models with a single ensemble member. As we focus on climate change signals, in line with IPCC practice, we do not apply bias correction as results are not expected to be substantially affected; but users interested in projecting impacts should use bias-corrected data from individual climate outcomes.

### Global warming levels

For each model ensemble member and target warming level, we selected the earliest 30-year window whose area-weighted global-mean temperature exceeds the area-weighted preindustrial (1851–1900) global-mean temperature by at least that target warming level. The same procedure was applied to the main time series used in the analyses—based on SSP5-8.5—and to those based on SSP2-4.5 used in Extended Data Fig. 11 to test the sensitivity of the results at 2 °C of global warming to the SSP. It is noted that the models used for analyses at different warming levels can vary because some models do not reach some of the highest warming levels.

### Identifying extreme climate outcomes

For each sector, once the global climatic impact-driver $f$ for climate outcomes is computed (equation (1)), see details below), models with $f$ values below and above the 8th and 92nd percentiles (for wildfires 12th and 88th percentiles given that fewer models are available for FWI) are selected as best and worst-case climate outcomes, respectively (Fig. 1); here and elsewhere, percentiles are computed with the R-function quantile with the recommended[60] algorithm type 8.

For the precipitation-related analysis, we derived highly populated areas in 2020 based on the GPWv4 population dataset[52]. We first obtained population density at each grid point (by dividing the population by the grid-point surface) and sorted grid points in terms of population density. Then, we derived the smallest land surface that includes 90% of the global population by selecting the first $N$ grid points whose aggregated total population covers 90% of the global population. Then, only grid points where no models (among the runs used for the 2 °C projections under SSP5-8.5) show an average Rx5day during the preindustrial period below 25 mm (5 mm $d^{-1}$) are retained. This avoids spuriously large percentage changes of Rx5day ($\Delta$Rx5day (%)) in very dry regions, which could disproportionately influence the global climatic impact-driver $f$ (it is noted that the worst-case model at 2 °C exceeds the multimodel mean at 3 °C also without this filtering). To compute $f$, for each model, we first re-gridded $\Delta$Rx5day (%) (the percentage difference in the time-mean of annual maxima consecutive 5-day precipitation between the future and the preindustrial periods) to the population dataset's grid via the nearest-neighbour approach and then computed the average (weighted by grid cell area) over the retained highly populated areas. It is noted that, in line with previous studies, we analyse local precipitation extremes as a proxy for local pluvial floods, assuming co-location and neglecting possible non-local flooding drivers.

For the drought analysis, we derived the total global breadbasket as the union of maize, wheat, soybean and rice breadbaskets from ref. 25. We computed $f$ for each model as the average (weighted by grid cell area) of $\Delta$Drought frequency (%) (the difference in the drought frequency between the future and the preindustrial periods) across grid points in the global breadbasket (where all models provide data). We defined grid-point droughts as annual average soil moisture low values occurring every 5 years on average (below the 20th percentile) in the preindustrial period. In Fig. 3a, Greenland, Antarctica and Iceland are excluded because these regions contain many locations for which not all models provide data.

For the FWI analysis, we used forest grid cells derived from the European Space Agency Land Cover dataset[54,55,61], which was first re-gridded to a 0.25° spatial grid using the LC-CCI user tool as in ref. 62. Then, forest grid cells were defined where the sum of tree densities (broadleaf and needleleaf, both evergreen and deciduous) exceeds the sum of the density of all other classes for all years during 2010–2019. It is noted that the worst-case models at 2 °C exceed the multimodel mean at 3 °C even with a less restrictive forest definition—namely, where the sum of tree densities exceeds 20% of the sum of the density of all classes (excluding water) for all years during 2010–2019. To compute $f$, for each model, we first re-gridded $\Delta$FWIx (the difference in the time-mean of annual maxima daily FWI between the future and the preindustrial periods) to the re-gridded land-cover dataset using the nearest-neighbour approach and then computed the spatially weighted average over forest grid cells.

For the analysis of temperature extremes on land, low-income countries were derived from ref. 32 as used by the IPCC[63]. We computed $f$ for each model as the average (weighted by grid cell area) of $\Delta$TXx (°C) (the difference in the time-mean of annual maxima daily maximum temperature between the future and the preindustrial periods) across grid points contained in the countries.

For the analysis of sea temperature extremes, we derived fisheries from the The Union of World Country Boundaries and EEZs (version 3)

dataset[64] as used in ref. 33 and removing polar regions. Building on ref. 33, which highlights the impact of future marine heatwaves on fish and fisheries[65], we defined $f$ as the average (weighted by grid cell area) of $\Delta$SSText (°C) (the difference in the 10-year return level—that is the 90th percentile—of the annual mean of sea surface temperature between the future and the preindustrial periods) across grid points contained in fisheries (where all models provide data).

### Coherent versus incoherent climate outcomes

For each sector, deriving extreme climate outcomes as described above implies that the worst-case outcome corresponds to the model $m^{\mathrm{w}}$ associated with the maximum value of $f_m$ (equation (1)). This yields a spatially coherent field, as it represents the projected climatic impact-driver from the same model at all locations $x$, that is $\Delta\mathrm{CID}_x^{(\mathrm{coh,w})} = \Delta\mathrm{CID}_{x,m^{\mathrm{w}}}$, where $m^{\mathrm{w}} = \arg\max_m f_m$. The corresponding $f$ value is:

$$f^{\mathrm{coh,w}} = \max_m \left( \frac{\sum_{x \in \text{critical area}} a_x\, \Delta\mathrm{CID}_{x,m}}{\sum_{x \in \text{critical area}} a_x} \right). \quad (2)$$

In contrast, in Extended Data Fig. 6, unrealistic, spatially incoherent worst cases are shown. Here the most extreme outcome assumes that each grid cell $x$ experiences the maximum projected climatic impact-driver across models, that is $\Delta\mathrm{CID}_x^{(\mathrm{incoh,w})} = \max_m \Delta\mathrm{CID}_{x,m}$, resulting in an unrealistic global field where different locations experience projections from different models. The corresponding $f$ value is:

$$f^{\mathrm{incoh,w}} = \frac{\sum_{x \in \text{critical area}} a_x\, (\max_m \Delta\mathrm{CID}_{x,m})}{\sum_{x \in \text{critical area}} a_x}. \quad (3)$$

The different placement of the max operator in $f^{\mathrm{coh,w}}$ and $f^{\mathrm{incoh,w}}$ illustrates the core problem of the incoherent approach, as applying the maximization before spatial averaging permits different models to determine different locations, undermining spatial consistency and inflating $f$ values. Likewise, the best-case outcome—obtained by replacing max with min—also breaks spatial consistency and deflates $f$ values under the incoherent approach. As a result of these inflated and deflated $f$ values, the incoherent approach exaggerates the uncertainty in $f$, defined as the difference between $f$ values for worst- and best-case climate outcomes (Extended Data Fig. 6c,f,i).

### Data availability

CMIP6 data can be retrieved at https://esgf-metagrid.cloud.dkrz.de/ search. Fire weather index data can be derived from ref. 53. Population data from the Gridded Population of the World (GPW), v4 dataset[52] were derived from https://sedac.ciesin.columbia.edu/data/set/ gpw-v4-population-count-rev11/data-download. Global breadbaskets can be derived from ref. 25 (a list of countries is available in their Supplementary Material). Forest grid cells were derived using the European Space Agency Land Cover dataset[54,55,61] (version 2.0.7 until 2015; version 2.1.1 from 2016) available at https://maps.elie.ucl.ac.be/CCI/viewer/ download.php, which was re-gridded using the LC-CCI user tool available at https://www.esa-landcover-cci.org/?q=node/163. Low-income countries were derived from Table 1 in ref. 32. Fisheries from the The Union of World Country Boundaries and EEZs (version 3) dataset[64] were derived from https://www.marineregions.org/downloads. php#unioneezcountry. Maps were plotted via the oce R-package[66]

and coastlines were generated using the maps R-package[67]. Source data are provided with this paper.

### Code availability

All codes are direct implementations of standard methods and techniques described in detail in Methods, and executed using standard Climate Data Operators (CDO) functions[68], R and Bash scripts. Code is available at https://zenodo.org/records/17864854 (ref. 69).

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

**Acknowledgements** E.B. received funding from the Deutsche Forschungsgemeinschaft (DFG, German Research Foundation) via the Emmy Noether Programme (grant ID 524780515). This project has received funding from the European Union's Horizon 2020 research and innovation programme under grant agreement number 101003469. J.Z. acknowledges the Helmholtz Initiative and Networking Fund (Young Investigator Group COMPOUNDX, Grant Agreement VH-NG-1537). We acknowledge A. Bastos and M. Anand for support with the re-gridding of the European Space Agency Land Cover dataset via the LC-CCI user tool, and F. Gaupp for making available the crop-region definitions. We acknowledge the World Climate Research Programme, which, through its Working Group on Coupled Modelling, coordinated and promoted CMIP6, and thank the climate modelling groups for producing and making available their model output, the Earth System Grid Federation (ESGF) for archiving the data and providing access, and the multiple funding agencies that support CMIP and the ESGF. Analyses were carried out on the high-performance computing cluster EVE, a joint effort of both the Helmholtz Centre for Environmental Research - UFZ and the German Centre for Integrative Biodiversity Research (iDiv) Halle-Jena-Leipzig.

**Author contributions** E.B. carried out the analyses, distilled the key result and wrote the paper. E.B., E.F. and J.Z. designed the global climatic impact-driver. E.B., E.F., J.S. and J.Z. discussed the results and participated in the review and editing of the paper.

**Funding** Open access funding provided by Helmholtz-Zentrum für Umweltforschung GmbH - UFZ.

**Competing interests** The authors declare no competing interests.

**Additional information**
**Correspondence and requests for materials** should be addressed to Emanuele Bevacqua.

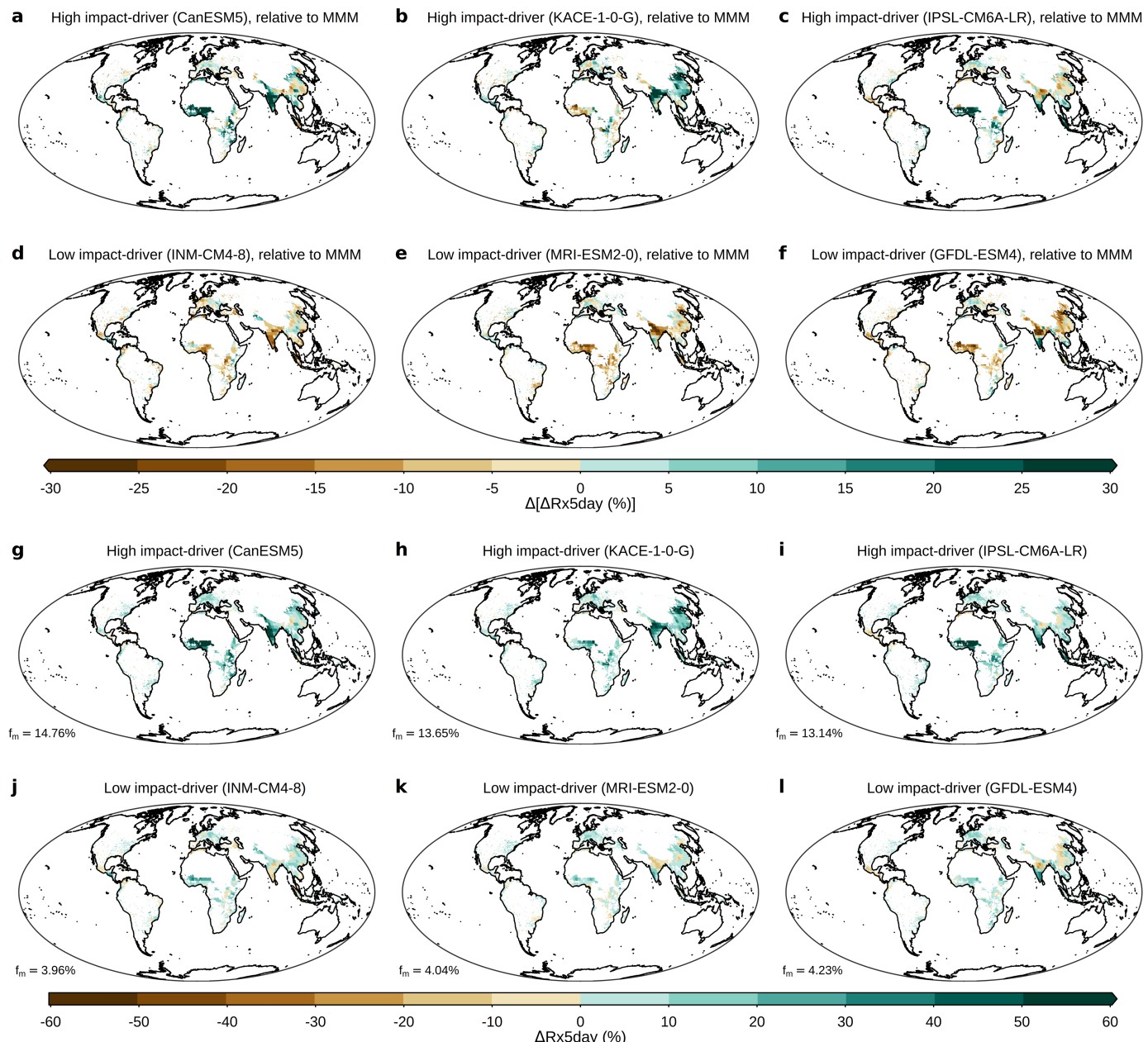

**a** High impact-driver (CanESM5), relative to MMM

**b** High impact-driver (KACE-1-0-G), relative to MMM

**c** High impact-driver (IPSL-CM6A-LR), relative to MMM

**d** Low impact-driver (INM-CM4-8), relative to MMM

**e** Low impact-driver (MRI-ESM2-0), relative to MMM

**f** Low impact-driver (GFDL-ESM4), relative to MMM

$\Delta[\Delta Rx5day (\%)]$

**g** High impact-driver (CanESM5)

$f_m = 14.76\%$

**h** High impact-driver (KACE-1-0-G)

$f_m = 13.65\%$

**i** High impact-driver (IPSL-CM6A-LR)

$f_m = 13.14\%$

**j** Low impact-driver (INM-CM4-8)

$f_m = 3.96\%$

**k** Low impact-driver (MRI-ESM2-0)

$f_m = 4.04\%$

**l** Low impact-driver (GFDL-ESM4)

$f_m = 4.23\%$

$\Delta Rx5day (\%)$

**Extended Data Fig. 1 | Extreme climate outcomes for precipitation extremes in highly populated areas from individual climate models in a 2 °C world. a-c** and **d-f**, $\Delta Rx5day$ (%) from individual 8% worst-case (a-c) and best-case (d-f) models in terms of the $f$ values used to identify extreme climate outcomes, shown as the difference with the multimodel mean (MMM). As a reference, note that Fig. 2c is based on the average of the worst-case models in panels a-c, and Fig. 2d is based on the average of the best-case models in panels d-f. **g-i** and **j-l**, The same as the first row (panels a-c) and second row (panel d-f), respectively, but they show the absolute value of $\Delta Rx5day$ (%) (no difference with multimodel mean).

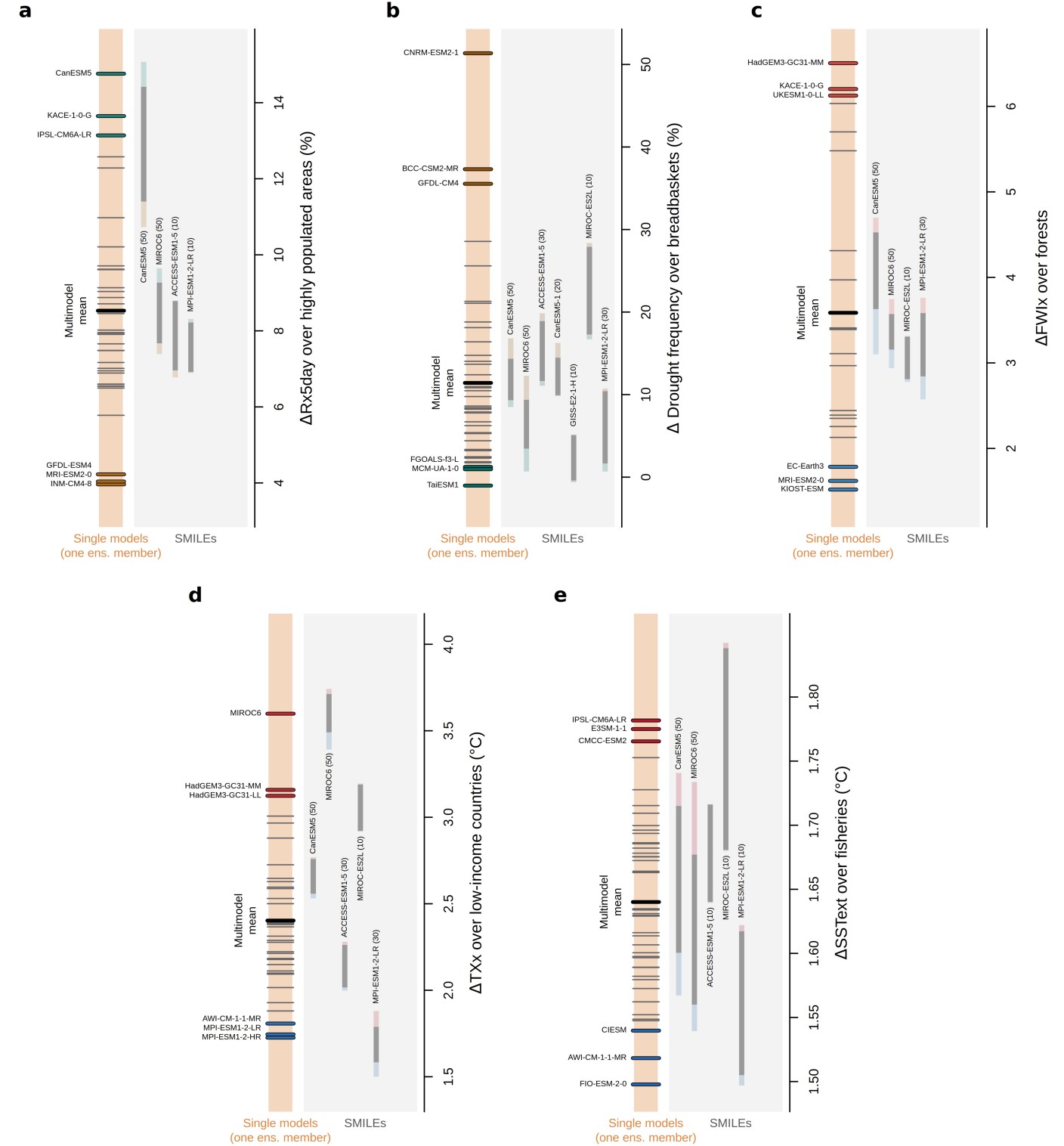

**Extended Data Fig. 2 | Global climatic impact-driver *f* from multiple Single Model Initial-condition Large Ensembles (SMILEs) in a 2 °C world. a**, For precipitation extremes in highly populated areas, the same as the left part of Fig. 2b, but with the addition of more SMILEs. Specifically, the left orange bar shows the global climatic impact-driver *f* used to identify climate outcomes computed for individual climate models in a 2 °C world (each model contributes with one ensemble member; the black thick line shows the multimodel mean). The models with the highest and lowest 8% values of *f* represent the worst and best cases (model names are provided). The multiple vertical lines to the right of the bar show the range of *f* for different SMILEs, with each SMILE labeled by its name and the number of ensemble members in brackets (the range of *f* is coloured for the highest and lowest 8% values derived from quantiles; note that for some SMILEs with only ten ensemble members, the 8% quantiles may nearly overlap with the minimum or maximum values). **b**, The same as panel a, but for droughts across breadbaskets worldwide. **c**, The same as panel a, but for fire weather extremes over forests; furthermore, highest and lowest values were defined via a 12% instead of an 8% threshold. **d**, The same as panel a, but for heat extremes in low-income countries. **e**, The same as panel a, but for marine heatwaves over fisheries. The *f* values of all SMILEs are provided in Extended Data Table 2.

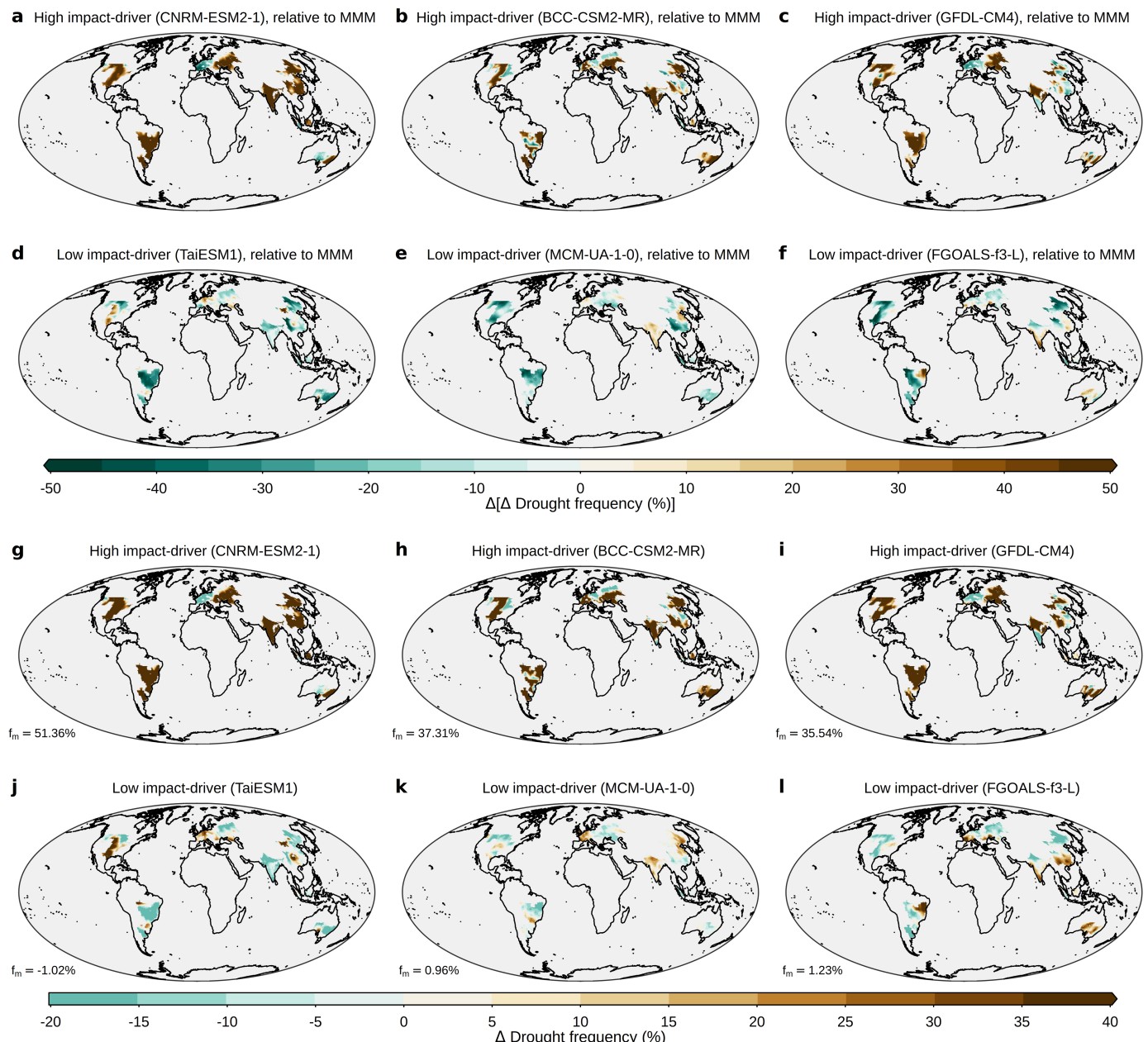

**a** High impact-driver (CNRM-ESM2-1), relative to MMM

**b** High impact-driver (BCC-CSM2-MR), relative to MMM

**c** High impact-driver (GFDL-CM4), relative to MMM

**d** Low impact-driver (TaiESM1), relative to MMM

**e** Low impact-driver (MCM-UA-1-0), relative to MMM

**f** Low impact-driver (FGOALS-f3-L), relative to MMM

Δ[Δ Drought frequency (%)]

**g** High impact-driver (CNRM-ESM2-1)

$f_m$ = 51.36%

**h** High impact-driver (BCC-CSM2-MR)

$f_m$ = 37.31%

**i** High impact-driver (GFDL-CM4)

$f_m$ = 35.54%

**j** Low impact-driver (TaiESM1)

$f_m$ = -1.02%

**k** Low impact-driver (MCM-UA-1-0)

$f_m$ = 0.96%

**l** Low impact-driver (FGOALS-f3-L)

$f_m$ = 1.23%

Δ Drought frequency (%)

**Extended Data Fig. 3 | Extreme climate outcomes for droughts across breadbaskets worldwide from individual climate models in a 2 °C world.** **a-c** and **d-f**, ΔDrought frequency from individual 8% worst-case (a-c) and best-case (d-f) models in terms of the *f* values used to identify extreme climate outcomes, shown as the difference with the multimodel mean (MMM). As a reference, note that Fig. 3c is based on the average of the worst-case models in panels a-c, and Fig. 3d is based on the average of the best-case models in panels d-f. **g-i** and **j-l**, The same as the first row (panels a-c) and second row (panel d-f), respectively, but they show the absolute value of ΔDrought frequency (no difference with multimodel mean).

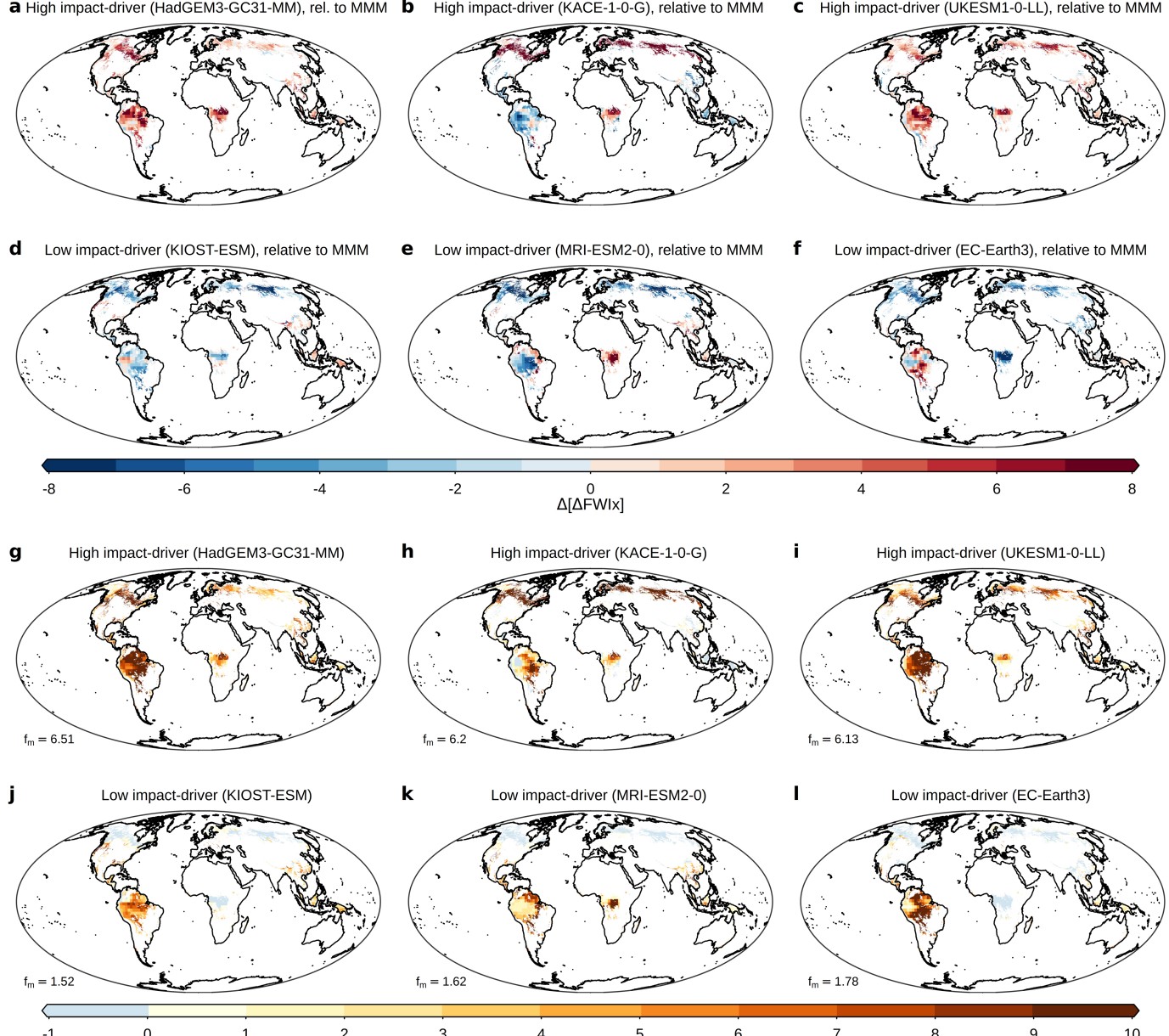

**Extended Data Fig. 4 | Extreme climate outcomes for fire weather extremes across forests worldwide from individual climate models in a 2 °C world. a-c** and **d-f**, ΔFWIx from individual 12% worst-case (a-c) and best-case (d-f) models in terms of the *f* values used to identify extreme climate outcomes, shown as the difference with the multimodel mean (MMM). As a reference, note that Fig. 4c is based on the average of the worst-case models in panels a-c, and Fig. 4d is based on the average of the best-case models in panels d-f. **g-i** and **j-l**, The same as the first row (panels a-c) and second row (panel d-f), respectively, but they show the absolute value of ΔFWIx (no difference with multimodel mean).

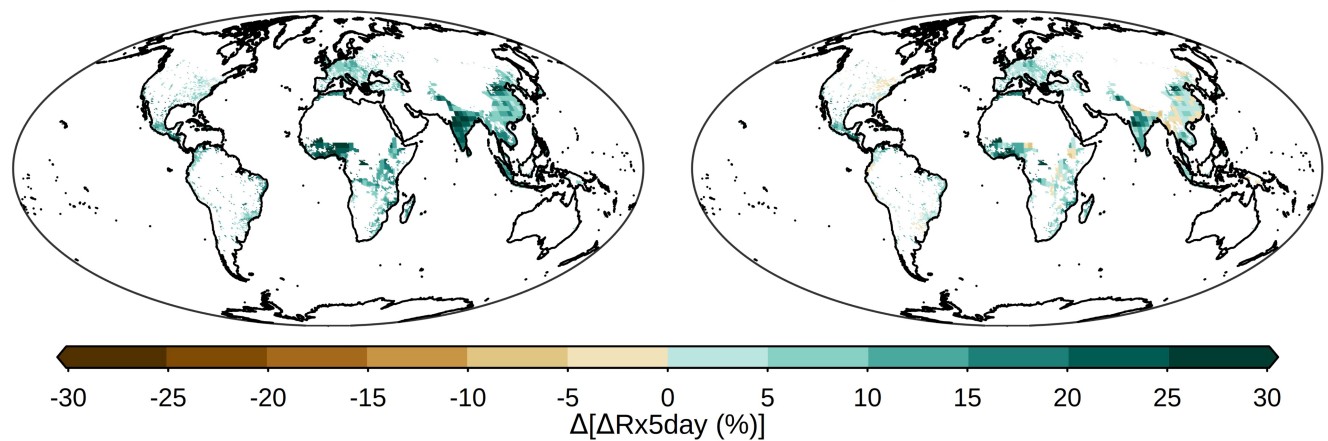

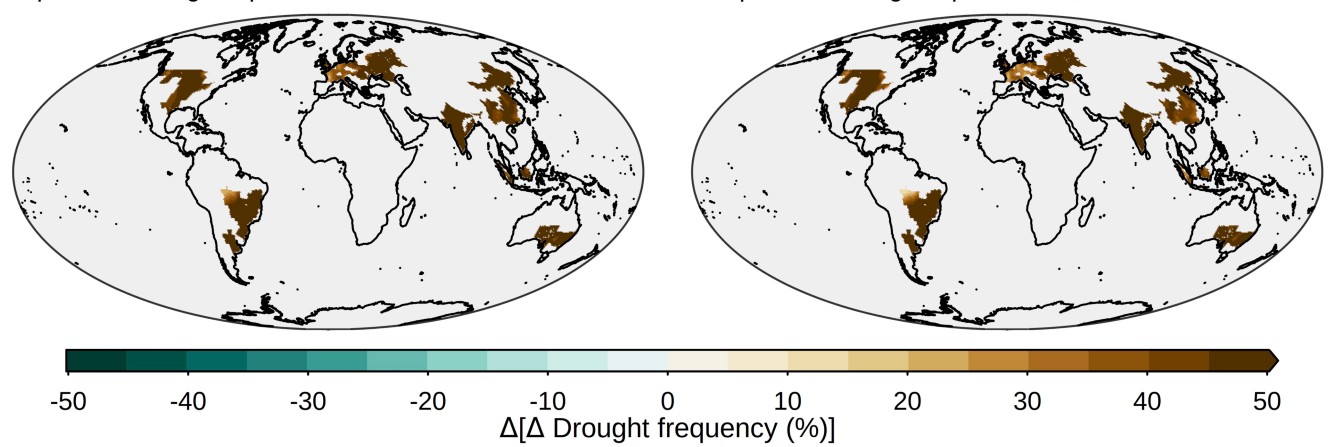

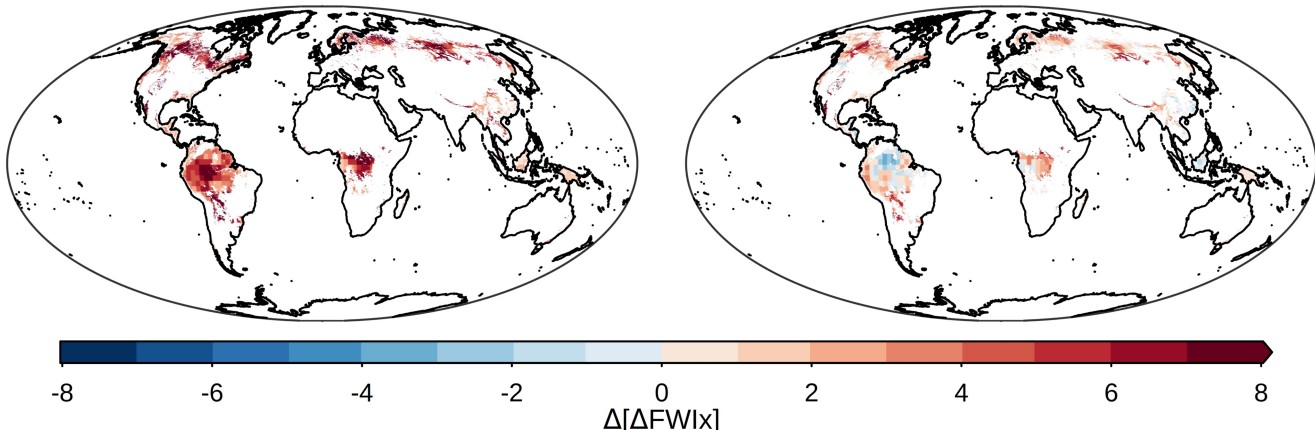

**Extended Data Fig. 5 | Deviation of spatially incoherent worst-case climate outcomes in a 2 °C world from multimodel mean at 3 °C and 4 °C. a**, For precipitation extremes in highly populated areas, the difference between (i) the local worst-case climate outcome in a 2 °C world (that is, at each individual location, the mean of ΔRx5day (%) across the 8% worst-case local ΔRx5day (%)) and (ii) the multimodel mean (MMM) in a 3 °C world. Note that for (i), deriving a global map this way yields a spatially incoherent climate outcome, as it ignores spatial dependencies by unrealistically assuming that different locations experience an outcome from a different set of models, whereas the multimodel

mean in (ii) follows standard practice by using the same, complete set of available models at all locations. **b**, The same as panel a, but the local worst-case climate outcome in a 2 °C world is compared against the multimodel mean in a 4 °C world. **c-d**, The same as panels a-b, but for droughts, thus for ΔDrought frequency across breadbaskets worldwide. **e-f**, The same as panels a-b, but for fire weather extremes, thus for ΔFWIx across forests; furthermore, worst and best cases were defined via a 12% instead of an 8% threshold. In all maps, consistent with the other analyses, both the local worst- and best-case climate outcomes are defined using three models.

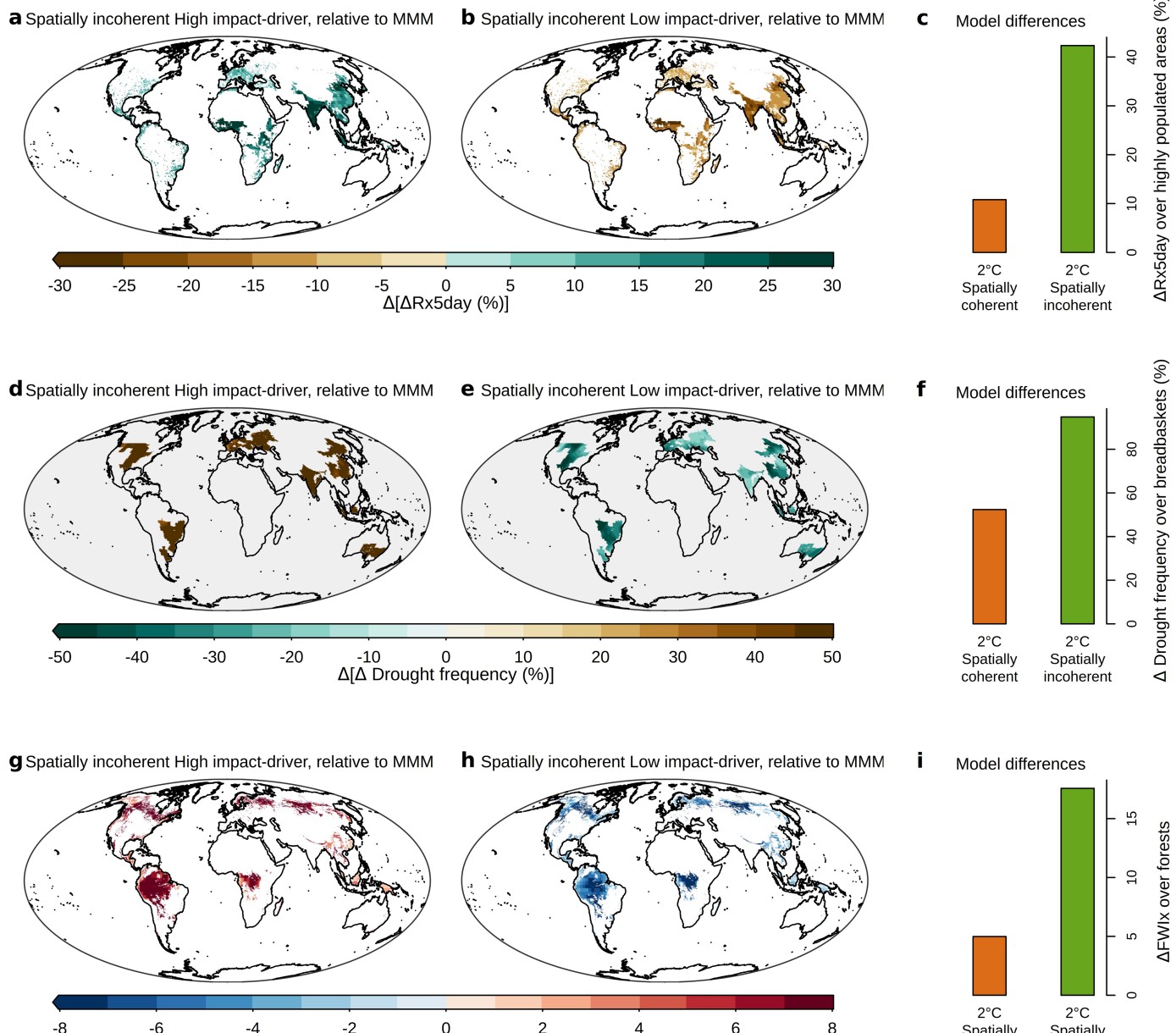

**a** Spatially incoherent High impact-driver, relative to MMM

**b** Spatially incoherent Low impact-driver, relative to MMM

**c** Model differences

Δ[ΔRx5day (%)]

**d** Spatially incoherent High impact-driver, relative to MMM

**e** Spatially incoherent Low impact-driver, relative to MMM

**f** Model differences

Δ[Δ Drought frequency (%)]

**g** Spatially incoherent High impact-driver, relative to MMM

**h** Spatially incoherent Low impact-driver, relative to MMM

**i** Model differences

Δ[ΔFWIx]

**Extended Data Fig. 6 | Spatially incoherent extreme climate outcomes in a 2 °C world relative to multimodel mean at 2 °C. a**, For precipitation extremes in highly populated areas, the same as Fig. 2c, but the global worst-case climate outcome in a 2 °C world is spatially incoherent. Specifically, it shows the difference between (i) the spatially incoherent global worst-case climate outcome in a 2 °C world built at each location independently as the mean of ΔRx5day (%) across the 8% worst-case local ΔRx5day (%) and (ii) the multimodel mean (MMM) in a 2 °C world. Note, the climate outcome in (i) is spatially incoherent as it is built ignoring spatial dependencies in ΔRx5day (%) by unrealistically assuming that different locations experience an outcome from a different set of models, whereas the multimodel mean in (ii) follows standard practice by using the same, complete set of available models at all locations. **b**, As panel a, but averaging at each location the 8% best ΔRx5day (%). **c**, Left orange bar: the range of *f* values across all models, which is identical to the left orange bar in Fig. 2e (this uncertainty range considers spatial dependencies in ΔRx5day (%)). Right green bar: the difference between the *f* values of two synthetic, unrealistic models that at each location take the worst- and best-case ΔRx5day (%) from all considered climate models (that is, unrealistic global worst- and best-case climate outcomes), respectively (see Methods). **d-f**, The same as panels a-c, but for droughts across breadbaskets worldwide, thus for ΔDrought frequency. **g-i**, The same as panels a-c, but for fire weather extremes over forests, thus for ΔFWIx; furthermore, in panels **g** and **h**, worst and best cases were defined via a 12% instead of an 8% threshold. In all maps, consistent with the other analyses, both the local worst- and best-case climate outcomes are defined using three models.

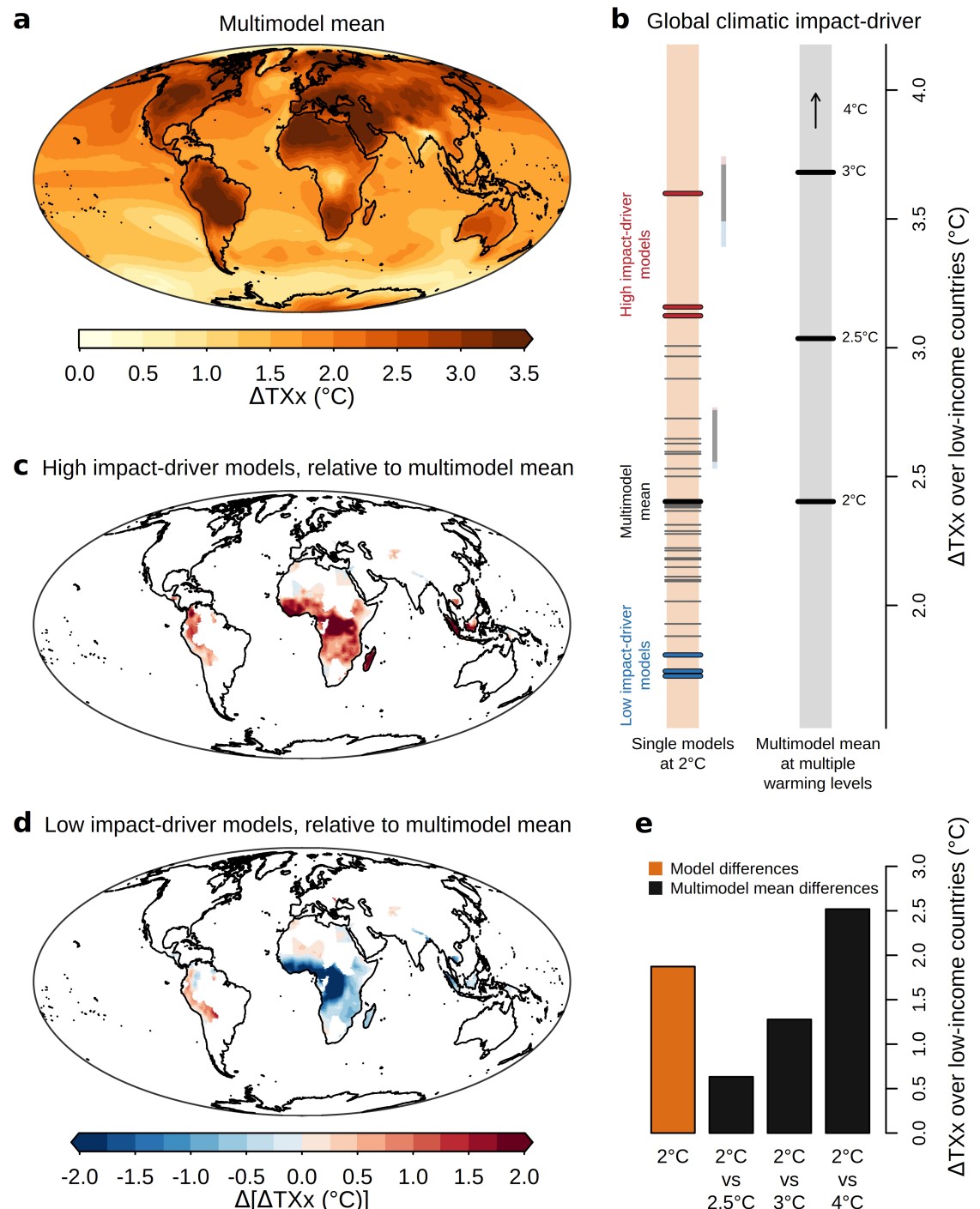

**Extended Data Fig. 7 | Extreme climate outcomes for heat extremes in low-income countries. a,** Multimodel mean of the projected mean annual maximum daily maximum temperature in a 2 °C warmer world relative to preindustrial conditions (1851–1900), that is, ΔTXx. **b,** Left orange bar: the global climatic impact-driver $f$ used to identify climate outcomes, that is, ΔTXx averaged over low-income countries (areas shown in panels c,d) ($f_m = \overline{\Delta TXx}_{m_{low-income\ countries}}$), computed for individual climate models in a 2 °C world (the black thick line shows the multimodel mean). The models with the highest and lowest 8% values of $f$ represent the worst and best cases (from top to bottom: MIROC6, HadGEM3-GC31-MM, and HadGEM3-GC31-LL (worst) and AWI-CM-1-1-MR, MPI-ESM1-2-LR, and MPI-ESM1-2-HR (best); see Extended Data Table 1 and Extended Data Fig. 2d

for model names). The two vertical lines to the right of the bar show the range of $f$ for two Single Model Initial-condition Large Ensembles (SMILEs; coloured for the highest and lowest 8% values – see Extended Data Fig. 2d for more SMILEs). Right bar: multimodel mean of $f$ at 2, 2.5, 3, and 4 °C global warming levels (the arrow indicates values above the y-axis range). **c,** Worst-case climate outcome at 2 °C, that is, difference between ΔTXx averaged across worst-case models and the multimodel mean, displayed over low-income countries. **d,** As panel c, but for the best-case outcome. **e,** Range of $f$ values across model simulations in a 2 °C world (left orange bar); and difference between multimodel mean of $f$ at 2 and 2.5 °C (second bar), 2 and 3 °C (third bar), and 2 and 4 °C (fourth bar). Data sources: climate models[51]; low-income countries[32].

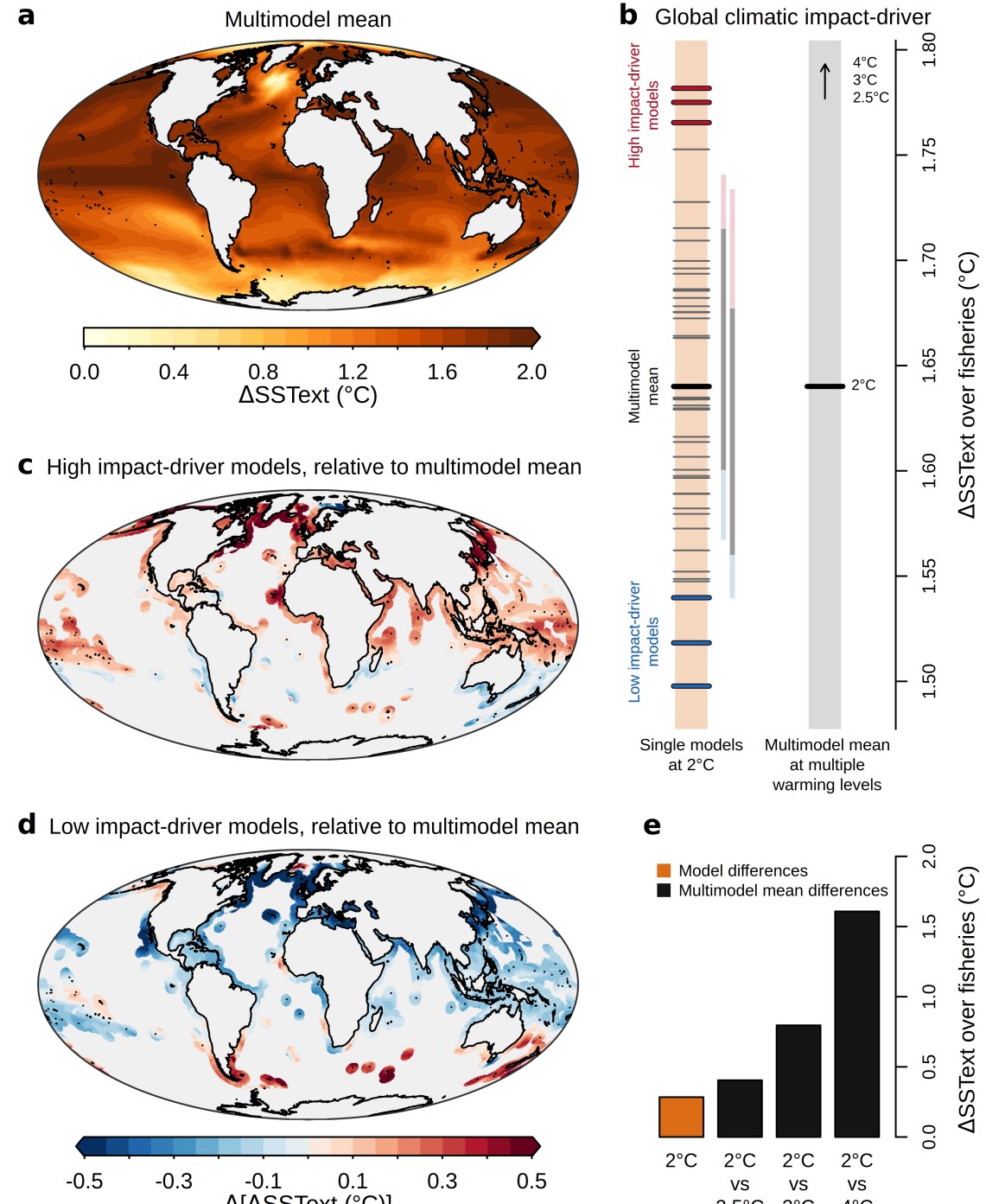

**Extended Data Fig. 8 | Extreme climate outcomes for marine heatwaves over fisheries. a**, Multimodel mean of the projected 10-year return level of sea surface temperature in a 2 °C warmer world relative to preindustrial conditions (1851–1900), that is, ΔSSText. **b**, Left orange bar: the global climatic impact-driver $f$ used to identify climate outcomes, that is, ΔSSText averaged over fisheries (areas shown in panels c,d) ($f_m = \overline{\Delta SSText_{m\,fisheries}}$), computed for individual climate models in a 2 °C world (the black thick line shows the multimodel mean). The models with the highest and lowest 8% values of $f$ represent the worst and best cases (from top to bottom: IPSL-CM6A-LR, E3SM-1-1, and CMCC-ESM2 (worst) and CIESM, AWI-CM-1-1-MR, and FIO-ESM-2-0 (best); see Extended Data Table 1 and Extended Data Fig. 2e for model names). Note that one model (horizontal grey line) lies almost entirely beneath the thick multimodel mean line. The two vertical lines to the right of the bar show the range of $f$ for two Single Model Initial-condition Large Ensembles (SMILEs; coloured for the highest and lowest 8% values − see Extended Data Fig. 2e for more SMILEs). Right bar: multimodel mean of $f$ at 2, 2.5, 3, and 4 °C global warming levels (the arrow indicates values above the y-axis range). **c**, Worst-case climate outcome at 2 °C, that is, difference between ΔSSText averaged across worst-case models and the multimodel mean, displayed over fisheries. **d**, As panel c, but for the best-case outcome. **e**, Range of $f$ values across model simulations in a 2 °C world (left orange bar); and difference between multimodel mean of $f$ at 2 and 2.5 °C (second bar), 2 and 3 °C (third bar), and 2 and 4 °C (fourth bar). Data sources: climate models[51]; fisheries[64].

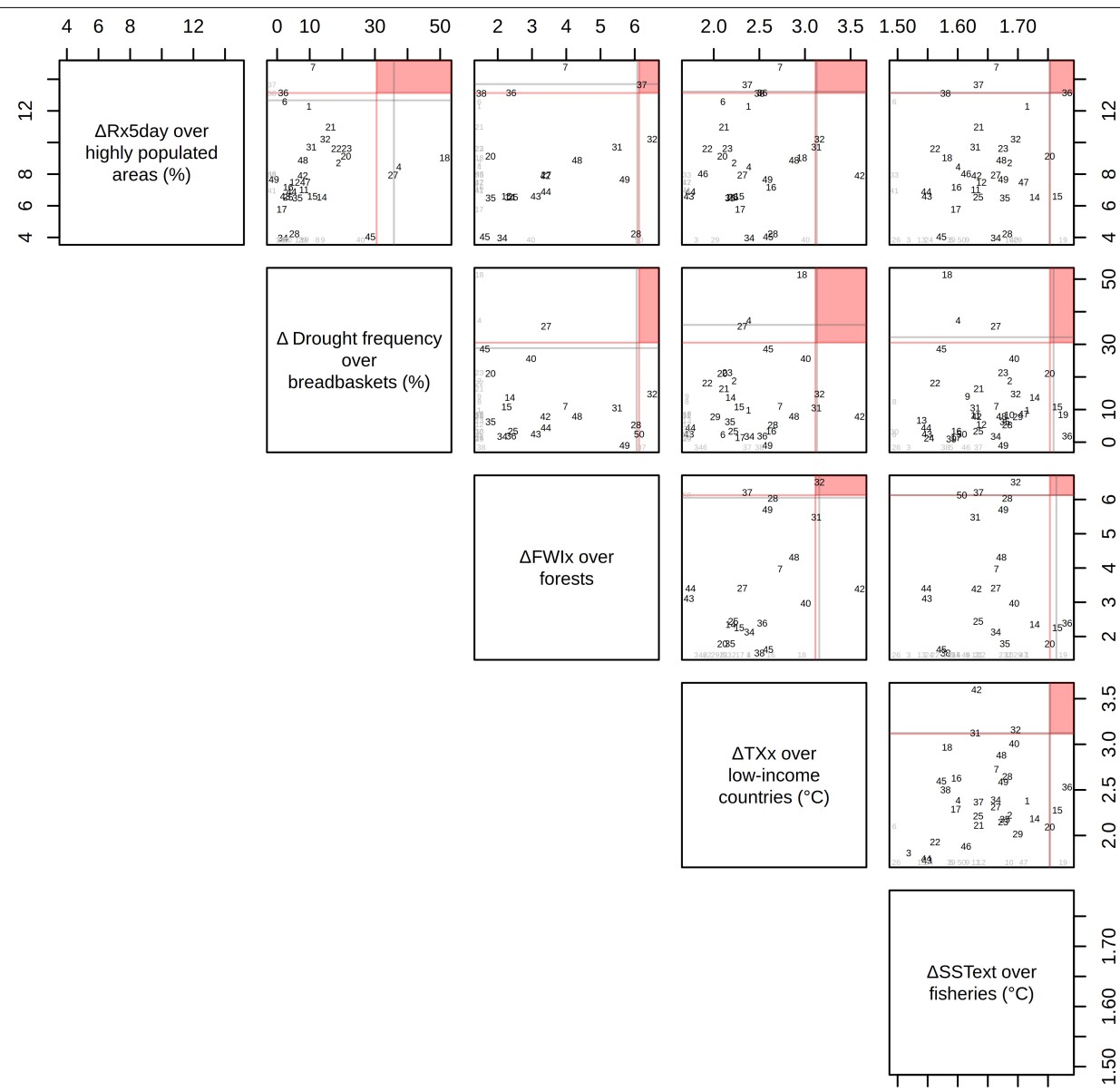

**Extended Data Fig. 9 | Correlations across sector-specific global climatic impact-drivers in a 2 °C world.** Scatterplots showing the relationship between global climatic impact-driver *f* values across the five considered sectors, with each panel comparing two sectors. Each model is identified by a number (following the numbering in the first column in Extended Data Table 1). Grey numbers along the x- or y-axis indicate *f* values for models that are available only for one of the two sectors. No significant Pearson correlations across models are found at the 5% level for any sector pairs. Red lines and the associated top-right corner mark the 92nd (88th for wildfires) percentile of *f*, calculated using all models available for the specific sector of interest (these percentiles are used to identify extreme climatic outcomes in the study). In contrast, grey lines show the percentiles calculated using only the subset of models simultaneously available for both sectors considered in each panel.

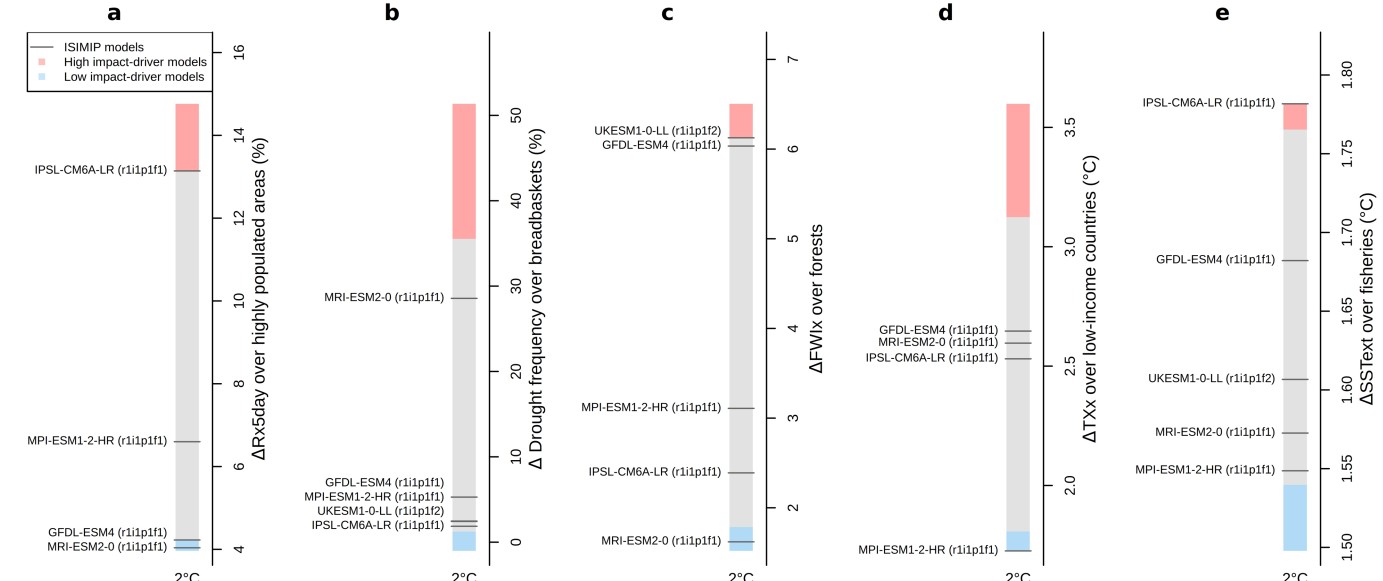

**Extended Data Fig. 10 | Positioning of the climate models of the official Inter-Sectoral Impact Model Intercomparison Project 3b (ISIMIP3b) protocol relative to the extreme climate models in a 2 °C world identified in this study. a-e**, For each of the five considered sectors, the shading illustrates the range of the values of the global climatic impact-driver *f* derived from all climate models considered in this study, with the range associated with the highest and lowest 8% (12% for fire weather over forests) models indicated in light red and blue, respectively. For reference, note that the range is as in panel b (left orange bar) of the main figure of each sector (Figs. 2–4 and Extended Data

Figs. 7, 8). Individual lines show the values of *f* for the models of the official ISIMIP3b protocol as described at https://protocol.isimip.org/#/ISIMIP3b/31-forcing-data (access date: 09/04/2025); note that for the ISIMIP sector of Fisheries and Marine Ecosystems, only output based on two climate models (GFDL-ESM4, IPSL-CM6A-LR) is available on the ISIMIP official repository at https://data.isimip.org/search/tree/ISIMIP3b/OutputData/ (access date: 09/04/2025). Note that for some of our sectors, the value of one or two of the five climate models employed in ISIMIP3b is not displayed as the climate model could not be considered in our analysis (Extended Data Table 1).

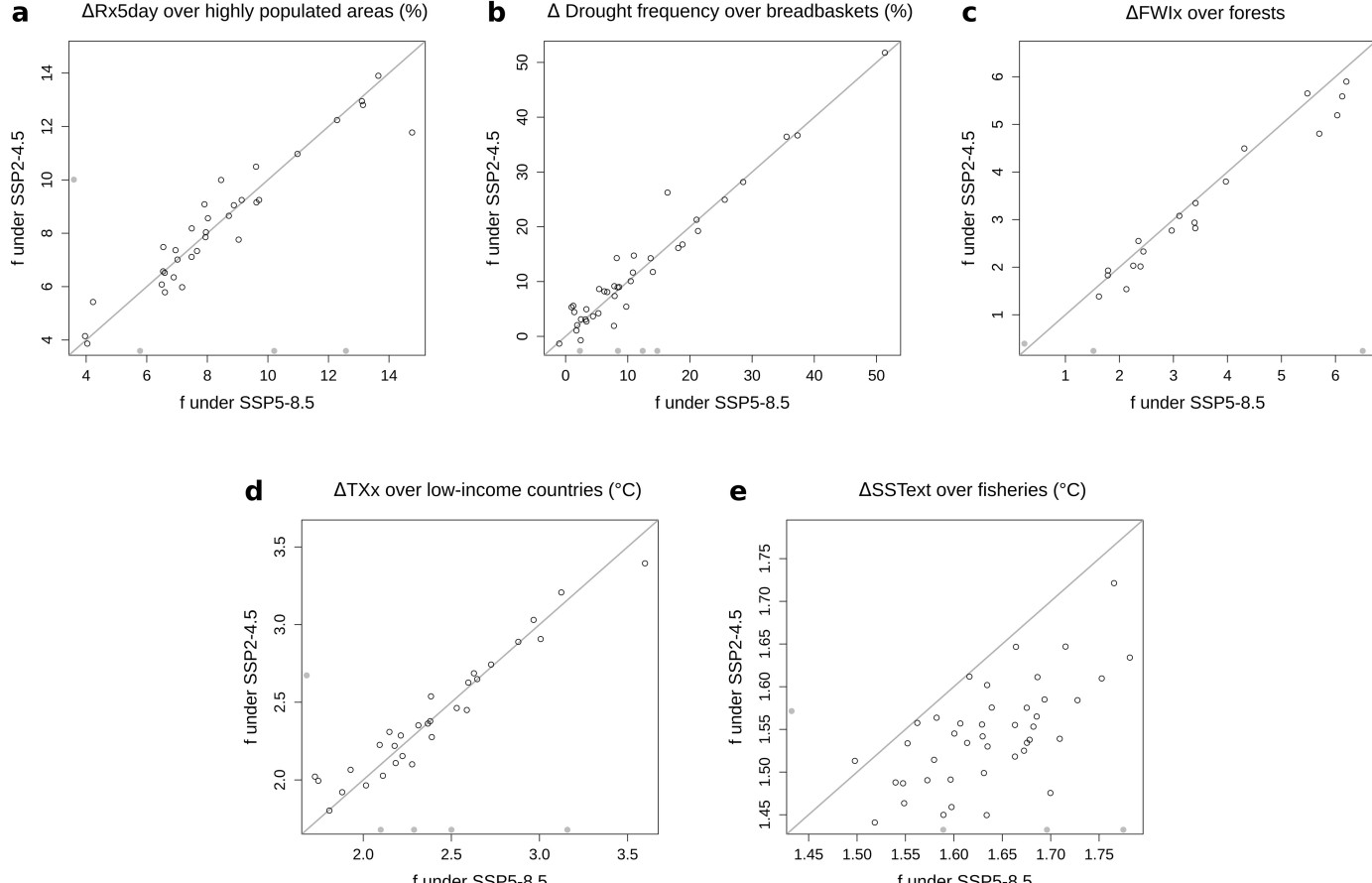

**Extended Data Fig. 11 | Sensitivity of extreme climate outcomes'
identification in a 2 °C world to using different SSP scenarios. a**, For
precipitation extremes in highly populated areas, scatterplot of the global
climatic impact-driver $f$ from different models obtained via (x-axis) the SSP5-8.5
scenario from 2015 onward (that is, as in Fig. 2b, left orange bar) and (y-axis) the
SSP2-4.5 scenario from 2015 onward. (Note that different ensemble members of
the same model may be used in the two scenarios, which does not compromise
comparability since differences due to internal climate variability are present
between SSP scenarios even when using ensemble members with the same ripf

index). Grey points along the x-axis and y-axis indicate the values of $f$ for models
that are available only for the SSP5-8.5 or SSP2-4.5 scenario, respectively. **b**, The
same as panel a, but for droughts across breadbaskets worldwide. **c**, The same
as panel a, but for fire weather index extremes across forests. **d**, The same as
panel a, but for heat extremes in low-income countries. **e**, The same as panel a,
but for marine heatwaves over fisheries; for this sector, note that differences in
f values between the two scenarios are small compared to the overall, average
f values. In general, some non-systematic differences between SSP scenarios
are expected due to internal climate variability.

**Extended Data Table 1 | Models used for the analyses in a 2 °C world and associated *f* values**

| Model (ensemble member) | Precipitation extremes in populated areas | Droughts over breadbaskets | Fire weather in forests | Heat in low-income countries | High oceanic temperature over fisheries |
|---|---|---|---|---|---|
| 1. ACCESS-CM2 (r1i1p1f1) | 12.28 | 9.76 | – | 2.38 | 1.72 |
| 2. ACCESS-ESM1-5 (r1i1p1f1) | 8.71 | 18.8 | – | 2.22 | 1.69 |
| 3. AWI-CM-1-1-MR (r1i1p1f1) | – | – | – | **1.81** | **1.52** |
| 4. BCC-CSM2-MR (r1i1p1f1) | 8.45 | **37.31** | – | 2.38 | 1.6 |
| 5. CAMS-CSM1-0 (r1i1p1f1) | – | – | – | – | 1.59 |
| 6. CAMS-CSM1-0 (r2i1p1f1) | 12.58 | 2.3 | – | 2.1 | – |
| 7. CanESM5 (r1i1p1f1) | **14.76** | 10.97 | 3.97 | 2.73 | 1.66 |
| 8. CanESM5-1 (r1i1p1f1) | – | 12.41 | – | – | – |
| 9. CanESM5-CanOE (r1i1p2f1) | – | 14.03 | – | – | 1.62 |
| 10. CAS-ESM2-0 (r1i1p1f1) | – | 8.37 | – | – | 1.69 |
| 11. CESM2 (r1i1p1f1) | 7.01 | 8.23 | – | – | 1.63 |
| 12. CESM2-WACCM (r1i1p1f1) | 7.48 | 5.39 | – | – | 1.64 |
| 13. CIESM (r1i1p1f1) | – | 6.7 | – | – | **1.54** |
| 14. CMCC-CM2-SR5 (r1i1p1f1) | 6.54 | 13.68 | 2.35 | 2.18 | 1.73 |
| 15. CMCC-ESM2 (r1i1p1f1) | 6.59 | 10.82 | 2.26 | 2.28 | **1.77** |
| 16. CNRM-CM6-1 (r1i1p1f2) | 7.17 | 3.36 | – | 2.63 | 1.6 |
| 17. CNRM-CM6-1-HR (r1i1p1f2) | 5.78 | 1.4 | – | 2.29 | 1.6 |
| 18. CNRM-ESM2-1 (r1i1p1f2) | 9.03 | **51.36** | – | 2.97 | 1.58 |
| 19. E3SM-1-1 (r1i1p1f1) | – | 8.42 | – | – | **1.78** |
| 20. EC-Earth3 (r1i1p1f1) | 9.13 | 21.07 | **1.78** | 2.09 | 1.75 |
| 21. EC-Earth3-CC (r1i1p1f1) | 10.98 | 16.39 | – | 2.11 | 1.63 |
| 22. EC-Earth3-Veg (r1i1p1f1) | 9.61 | 18.12 | – | 1.93 | 1.56 |
| 23. EC-Earth3-Veg-LR (r1i1p1f1) | 9.62 | 21.3 | – | 2.15 | 1.68 |
| 24. FGOALS-f3-L (r1i1p1f1) | – | **1.23** | – | – | 1.55 |
| 25. FGOALS-g3 (r1i1p1f1) | 6.55 | 3.34 | 2.44 | 2.21 | 1.63 |
| 26. FIO-ESM-2-0 (r1i1p1f1) | – | – | – | – | **1.5** |
| 27. GFDL-CM4 (r1i1p1f1) | 7.94 | **35.54** | 3.41 | 2.31 | 1.66 |
| 28. GFDL-ESM4 (r1i1p1f1) | **4.23** | 5.28 | 6.03 | 2.65 | 1.68 |
| 29. GISS-E2-1-G (r1i1p1f2) | – | 7.78 | – | 2.02 | 1.7 |
| 30. GISS-E2-1-H (r1i1p1f2) | – | 3.21 | – | – | – |
| 31. HadGEM3-GC31-LL (r1i1p1f3) | 9.71 | 10.47 | 5.48 | **3.12** | 1.63 |
| 32. HadGEM3-GC31-MM (r1i1p1f3) | 10.21 | 14.76 | **6.51** | **3.16** | 1.7 |
| 33. IITM-ESM (r1i1p1f1) | 7.95 | – | – | – | – |
| 34. INM-CM4-8 (r1i1p1f1) | **3.96** | 1.74 | 2.13 | 2.39 | 1.66 |
| 35. INM-CM5-0 (r1i1p1f1) | 6.5 | 6.24 | 1.79 | 2.18 | 1.68 |
| 36. IPSL-CM6A-LR (r1i1p1f1) | **13.14** | 1.86 | 2.39 | 2.53 | **1.78** |
| 37. KACE-1-0-G (r1i1p1f1) | **13.65** | – | **6.2** | 2.37 | 1.63 |
| 38. KIOST-ESM (r1i1p1f1) | 13.11 | – | **1.52** | 2.5 | 1.58 |
| 39. MCM-UA-1-0 (r1i1p1f2) | – | **0.96** | – | – | 1.59 |
| 40. MIROC-ES2L (r10i1p1f2) | – | 25.6 | 2.97 | 3.01 | 1.69 |
| 41. MIROC-ES2L (r1i1p1f2) | 6.95 | – | – | – | – |
| 42. MIROC6 (r1i1p1f1) | 7.91 | 7.84 | 3.39 | **3.6** | 1.63 |
| 43. MPI-ESM1-2-HR (r1i1p1f1) | 6.6 | 2.47 | 3.11 | **1.73** | 1.55 |
| 44. MPI-ESM1-2-LR (r1i1p1f1) | 6.89 | 4.41 | 3.4 | **1.74** | 1.55 |
| 45. MRI-ESM2-0 (r1i1p1f1) | **4.04** | 28.56 | **1.62** | 2.6 | 1.57 |
| 46. NESM3 (r1i1p1f1) | 8.02 | – | – | 1.88 | 1.61 |
| 47. NorESM2-LM (r1i1p1f1) | 7.49 | 8.61 | – | – | 1.71 |
| 48. NorESM2-MM (r1i1p1f1) | 8.88 | 7.89 | 4.31 | 2.88 | 1.67 |
| 49. TaiESM1 (r1i1p1f1) | 7.66 | **-1.02** | 5.7 | 2.59 | 1.68 |
| 50. UKESM1-0-LL (r1i1p1f2) | – | 2.43 | **6.13** | – | 1.61 |
| Total number of employed models | 36 | 42 | 22 | 34 | 45 |

| | | | | | | | | | |
|---|---|---|---|---|---|---|---|---|---|
| 0–10% | 10–20% | 20–30% | 30–40% | 40–50% | 50–60% | 60–70% | 70–80% | 80–90% | 90–100% |

For a given sector, the hyphen '–' indicates that the model (ensemble member, uniquely identified by the standard CMIP6 ripf index) was not employed. The values are the global climatic impact-driver *f* in panel b of each sector's main figure, with the highest and lowest *f* values—corresponding to the worst- and best-case models—underlined and in bold. For each sector, cell colour indicates the model's rank based on deciles derived from available models (see legend). Note that for CAMS-CSM1-0 and MIROC-ES2L, different ensemble members were used across sectors. Extended Data Fig. 9 shows a scatterplot of *f* values across sectors.

**Extended Data Table 2 | Single Model Initial-condition Large Ensembles (SMILEs) used for the analyses in a 2 °C world and associated *f* values**

| Model (ensemble member) | Precipitation extremes in populated areas | Droughts over breadbaskets | Fire weather in forests | Heat in low-income countries | High oceanic temperature over fisheries |
|---|---|---|---|---|---|
| CanESM5 (r1–25(i1p1f1), r1–25(i1p2f1)) | 12.96 (15.07, 10.72) | 11.92 (16.82, 8.5) | 4.03 (4.7, 3.1) | 2.66 (2.77, 2.53) | 1.65 (1.74, 1.57) |
| MIROC6 (r1–50(i1p1f1)) | 8.5 (9.65, 7.39) | 6.22 (12.29, 0.69) | 3.37 (3.75, 2.94) | 3.61 (3.74, 3.39) | 1.63 (1.73, 1.54) |
| ACCESS-ESM1-5 (r1–30(i1p1f1)) | – | 15.06 (19.86, 11.07) | – | 2.14 (2.28, 2) | – |
| ACCESS-ESM1-5 (r1–10(i1p1f1)) | 8.31 (8.8, 6.78) | – | – | – | 1.68 (1.72, 1.64) |
| CanESM5-1 (r1–10(i1p1f1), r1–10(i1p2f1)) | – | 12.31 (16.27, 9.83) | – | – | – |
| GISS-E2-1-H (r1–5(i1p1f2), r1–5(i1p3f1)) | – | 2.27 (5.17, -0.62) | – | – | – |
| MIROC-ES2L (r1–10(i1p1f2)) | – | 22.93 (28.36, 16.7) | 3.02 (3.31, 2.78) | 3.05 (3.19, 2.92) | 1.75 (1.84, 1.68) |
| MPI-ESM1-2-LR (r1–30(i1p1f1)) | – | 6.46 (10.72, 0.7) | 3.17 (3.76, 2.57) | 1.69 (1.88, 1.5) | – |
| MPI-ESM1-2-LR (r1–10(i1p1f1)) | 7.49 (8.31, 6.89) | – | – | – | 1.57 (1.62, 1.5) |

Some models appear in multiple rows because different ensemble members (uniquely identified by the standard CMIP6 ripf index) were used across sectors. Specifically, for a given sector, the hyphen '–' indicates that the model was either not employed or not analysed under the same ensemble member configuration as in other sectors. The reported statistics are the mean (max, min) *f* values across ensemble members. The statistics of the first two SMILEs (CanESM5 and MIROC6) are plotted to the right of the orange bar in panel b of each sector's main figure, while all SMILEs are plotted to the right of the orange bar in Extended Data Fig. 2.