## [Peer Review file · Nature]

Moderate global warming does not rule out extreme global climate outcomes

Corresponding Author: Dr Emanuele Bevacqua

Version 0:

Reviewer comments:

Referee #1

(Remarks to the Author)

Review of 'Moderate global warming does not rule out extreme global climate outcomes' by Bevacqua et al

This paper considers the best and worst case outcomes at 2degrees of global warming across a range of climate models, highlighting the strong structural model uncertainty in the variables considered. This paper is important for our understanding and communication of possible climate impacts across multiple sectors and provides an important piece of literature. I recommend publishing with minor revisions stated below.

Model independence and model realism are not considered in the paper: there is a brief discussion at the end but I think this could be important and it is why my following suggestions ask for model names on some of the plots to Figure out which model is behaving in which way. Here is one example of a paper that discussed these important factors:
<https://esd.copernicus.org/articles/11/995/2020/>

The two large ensembles have quite different ranges of internal variability. Many more ensembles exist. Why did you choose this specific two? Would it be worth adding another few to see how they compare? This leads back to my first question around model performance – are some models better than others? Additionally if it is worth adding a few lines to the paper discussing how the worst or best case model could be even worse or better if the member selected happens to be an end member on the internal variability spectrum. For the current list of available ensembles see here:
<https://egusphere.copernicus.org/preprints/2024/egusphere-2024-3684/>

Figures in General:

For Figure 2 and associated similar Figs.

Which large ensemble is which on the plots?

Have you considered putting the names of the end member models on panel b) so the reader knows which model is which? I find it confusing that c and d only show subsets of a – can you put dots or lines or find some way to show which regions are in both a c and d?

Figure captions: I would find it easier if the important parts were in each caption rather than 'as in Fig 2' otherwise the reader has to keep flipping back and forth for the important details

Line 145 – does it make sense to use this fire forest index everywhere? It seems odd there is no data in Australia where we know there are fires?

SSP370 is quite a different pathway to SSP585 can you please redo your comparison in Ext Fig 10 with SSP370

Code availability – this is not sufficient to reproduce the work done in the paper

Ext Table 1: I recommend putting in brackets the min/mean or medium/max values across the 2 large ensembles. For CanESM5 anecdotally the p1 and p2 ensembles are quite different. Please check that if you split into these two halves that the statistics are the same. One more suggestion is to add a colorbar to the table and highlight models in ranks blue could be 1: red could be the last one. This is often done for evaluation and would show if some models are always to one side of the distribution. An example of what I mean is here: https://pcmdi.llnl.gov/pmp-preliminary-results/interactive_plot/portrait_plot/enso_metric/enso_metrics_interactive_portrait_plots_v20231121.html

Ext Figs 1-5: Is there a better projection you can use as much of the map is currently white space?

Ext Fig 4-5 is the MMM also spatially incoherent in these plots? I was a little confused about what was exactly being plotted. Please check the captions.

Ext Fig 8: Perhaps replace dots with model numbers? And number the models in Ext Table 1? This would help see model coherency on the plot.

Minor details:

Line 47 please reword 'besides others' and be more specific

Line 75 replace 'other' with 'another'

Line 78 – please define Pluvial

Line 172-186 – I found these sentences hard to follow can you read and revise please

Line 232 – please refer to the Table so we know where to find the model list – same in Data Availability

In the methods I couldn't find how the warming levels were calculated

Referee #2

(Remarks to the Author)

SUMMARY OF KEY RESULTS, ORIGINALITY AND SIGNIFICANCE

This is a potentially agenda-setting paper which provides an original take on a 'safe landing' approach to climate risk management. Instead of focusing on a 'cascade of uncertainty' when projecting multi model means and ranges for global climate risks, or indeed on the constraints which would avoid certain undesirable outcomes for global climate change impacts (such as the collapse of ocean currents) the authors focus on an analysis of the potential for extreme outcomes in areas of global importance for the survival of humans and ecosystems respectively. Specifically, some key drivers of climate change impacts that are produced by GCMs in the CMIP6 ensemble dataset are selected and worst case outcomes for each climate model, over these specific areas (which are not necessarily spatially contiguous). By showing that we cannot rule out adverse outcomes for low levels of warming such as 2C, they provide vital new evidence relevant to decision making. The policy relevance of this cannot be underestimated, especially because the authors focus on three local outcomes that are globally relevant, by aggregating local outcomes for (a) heavy 5 day precipitation events in areas where 90% of the human population lives (b) soil moisture deficit in grid cells that form the global breadbasket and (c) enhancement of fire weather in ecologically critical forests.

The findings are alarming: they demonstrate that across multiple sectors, the climate outcomes projected for a 2C world, include a significant number of worse-case scenarios (around 10%) in which the outcomes EXCEED SIGNIFICANTLY the outcomes associated with the multi-model mean outcomes for higher levels of warming. This suggests that a precautionary approach is needed to avoid these outcomes with greater than 90% probability, of limiting level of warming to well below 2C as stipulated in the Paris Agreement. Specifically, the 8% worse case projections of heavy 5 day precipitation in areas where 90% of people live exceeds the multi-model mean for 3C warming; and the four worst case models quantifying exposure of forests to fire weather at 2C warming exceed the multi-model projections at 3C warming. Uncertainty in model projection of soil moisture deficit is found to be very large with a quarter of models projecting outcomes at 2C which exceed the multi model mean at 4C warming.

The originality and policy relevance of the approach convinces me that this paper is appropriate for publication in Nature. The implication of the paper is a far greater need for climate change mitigation in order to avoid potentially serious outcomes for humans and ecosystems with high confidence.

The paper is potentially an agenda setting research paper, since other climate change impact models (with for example models of flood and drought risk modelled hydrologically, models of biodiversity loss and so on) could utilize this safe landing approach to the subsets of the globe where adverse impacts would be particularly important to avoid.

DATA/METHODS: VALIDITY, QUALITY OF DATA, QUALITY OF PRESENTATION, STATISTICS AND TREATMENT OF UNCERTAINTIES

The methods and results are very clearly presented. The data sets selected are widely accepted, publicly available on well known reliable websites of established institutions and organisations that curate and archive climate model output and population data. The quality of the data utilised is state of the art.

The analysis is sound, although I suggest some improvements/clarifications as follows:

- (1) Line 66 states that 10% is used at the cutoff for extreme values of the hazard indicator f . However it is later explained that an 8% cutoff is used for heavy precipitation events and 12% for fire weather. Explain/correct inconsistency.
- (2) In the methods section, a number of acronyms are used to describe ensemble members - these acronyms need to be detailed. Later in extended data table 1, 'rlilp1f1' and other character strings require explanation.
- (3) Why are the first ensemble members chosen for the analysis? Is this simply meant to be a 'random' selection? Is there any reason why that might not be the case?

(4) Why is the full ensemble only studied for two GCMs and why those particular two? Are the full ensembles available for the other GCMs, or only for those two? If the scope of the study meant that only two could be included, then it is important to justify why these two in particular were selected.

Extended figure 5 is particularly useful in showing what would happen if the approach were (incorrectly) applied to everywhere (in each grid cell) simultaneously without considering spatial coherence. It would be helpful to detail mathematically, the formula used to construct this to show how it differs from the formula used to construct figure 2 which is spatially coherent, to make this very clear to a wider range of readers. This is quite critical to convincing the reader that the approach is mathematically sound (by illustrating how NOT to do it ... the key advance made by the authors is in fact figuring out an alternative to this 'wrong' way of proceeding, and it is why this has not, to my knowledge, been done before.

It would also be interesting to create a comparison, in Figure 5, as follows: what would be the outcome if spatial coherence was accounted for, but the metrics are averaged over the whole global land surface for each climate model and then the worst 10% are extracted? This would show whether the areas where people live and grow crops are disproportionately exposed to climate hazard as compared with the whole of the terrestrial land.

Extended figure 8 is difficult to understand. Please elaborate the figure caption.

Extended figure 10 is useful, but the text explaining how it is derived (line 235) requires more explanation to document how the time slices or years at which 2C warming is reached are extracted from the two time series for the two SSPs.

CONCLUSIONS: ROBUSTNESS, VALIDITY, RELIABILITY

The conclusions are valid, and the implications of the paper, in terms of the methodological insights and advance it provides, would be expected to be robust to the selection of different sets of GCM ensemble members.

The exact numbers might be affected by the choice of the two particular GCMs ensembles, hence my request above that the authors detail whether there was any particular reason for this choice.

SUGGESTED IMPROVEMENTS, FOR POSSIBLE REVISION

Given the size of the ISIMIP community, consider moving Figure 9 into the main text.

The authors should acknowledge more fully the gap between a single measure of heavy precipitation and a flood risk outcome; similarly the gap between a soil moisture deficit and a crop failure. I suggest that the authors modify the wording in the paper to explain that they are quantifying exposure to climatic hazard, rather than the risk of an adverse outcome itself.

The vulnerability of the human or natural systems is affected by physical processes mediating the response to moisture deficit or excess, for example, soil texture or river flows between grids. Beyond this, risk itself is of course mediated by the vulnerability and potential to adapt to the changes. This is of course beyond the scope of the paper, and I am not suggesting that the authors should extend their calculations to cover these aspects, but they should acknowledge them in more detail.

The authors specifically single out the ISIMIP protocol, including its basis on 5 single GCMs, to show the inadequacy of the approach in the light of this analysis. However, the authors also argue that the approach should be applied by risk assessment studies more generally. For example, a flood risk model creating hydrological projections across CMIP6 ensemble could instead focus on these heavily populated regions and explore these extreme outcomes. The authors might go a little further in recommending how their approach might be applied to 'climate impact models', the running of which create many of the projects used in IPCC assessment reports, which have tended to focus on global or regional scale analysis. A more sectoral based approach is also appropriate for adaptation studies – these are the worst outcomes which may need to be planned for.

The authors would do well to explain that their approach is important for informing both climate change mitigation policy and also climate change adaptation policy.

REFERENCES – APPROPRIATE CREDIT TO PREVIOUS WORK?

An extended reference list provides context. However it would be helpful to cite the literature on climate risk hotspots, eg Byers, E., et al., 2018: Global exposure and vulnerability to multi-sector development and climate change hotspots. *Environ. Res. Lett.*, 13(5), 55012, doi:10.1088/1748-9326/aabf45. It would also be useful to cite some of the literature which projects climate risk that are based largely on multi model means, e.g. Arnell, N.W., J.A. Lowe, B. Lloyd-Hughes and T.J. Osborn, 2018: The impacts avoided with a 1.5°C climate target: a global and regional assessment. *Clim. Change*, 147(1), 61–76, doi:10.1007/s10584-017-2115-9; Warren, R., Andrews, O., Brown, S., et al. 2022. Quantifying implications of limiting global warming to 1.5 or 2°C above pre-industrial levels. *Climatic Change* 172: 39

CLARITY AND CONTEXT, LUCIDITY OF ABSTRACT/SUMMARY, APPROPRIATENESS OF ABSTRACT, INTRODUCTION AND CONCLUSIONS

The paper is generally extremely well written, and is in the style appropriate for publication in *Nature*. The abstract and introduction are incisive and easy to understand for the general reader. The conclusions are appropriate, but could be extended as recommended above.

Version 1:

Reviewer comments:

Referee #1

(Remarks to the Author)

The authors have responded extensively to all my comments, and I feel the paper can be accepted as is.

Referee #2

(Remarks to the Author)

I am now very happy to recommend this revised paper for publication as is. It was truly a pleasure to read the careful responses the authors have written to my comments. I was particularly impressed with (1) the way they handled my

suggestion to use mathematical formulas to distinguish between the method that they applied and to compare it with a method that would instead be misleading (2) the way that they added additional 'SMILES' to Extended Figure 2 to demonstrate that inter-model variability exceeds internal variability even if additional SMILES are incorporated and (3) that they took the trouble to conduct an additional analysis to show the implications of a coherent calculation of risk indicators in which the average across the land surface is taken before the worst outcomes are extracted. I was also grateful for the explanation of the CMIP6 acronyms which convinced me that their sampling procedure is indeed truly random. Finally, I felt that they handled my comment concerning the use of terms 'risk' 'hazard' and 'exposure' well and made appropriate modifications to the text in a number of important places. I also feel that the authors have taken up well my suggestion to enhance the discussion of the implications of their methods in informing climate change risk analysis going forward.

Response to Referees

We thank the referees and the editor for the time spent reviewing the paper. Comments and suggestions were very valuable and constructive, and they have contributed to improving the manuscript. We have addressed the individual comments and carefully revised the manuscript accordingly. A new manuscript version is attached.

Please find our response (shown in regular text) to the individual comments of the reviewers (in blue and italics) below. When showing changes to the paper, new text is indicated in bold (note that, for simplicity, references are denoted as *[Reference]*, with the exact reference numbers visible in both the tracked-changes and PDF versions).

Figures in this file are named as in the paper when identical to those in the manuscript; otherwise, they are labelled as Figure R1, R2, and R3.

Due to the addition of a new Extended Data Figure, the numbering of the Extended Data Figures has changed. In our replies, we consistently use the updated numbering, and when a reviewer refers to an older figure number, we clarify discrepancies to avoid misunderstandings.

In addition to the changes in response to the comments, here we note that:

- 1) We slightly modified the last sentence of the current third last paragraph of the discussion: *Old version*: “For instance, multiple drought-sensitive sectors are likely to experience a worst-case dry outcome concurrently.” → *New version*: “For instance, should a worst-case dry outcome occur, multiple drought-sensitive sectors would likely be affected simultaneously.”
- 2) We modified a sentence in the abstract by adding a few words (in bold here): “worst-case global climates typically communicated via **the average of** climate model projections at high global warming levels”.
- 3) We improved the spatial mask used to define low-income countries (for the analysis of heatwaves shown in the Extended Data), to include Indonesia, and remove W. Sahara from the Morocco’s polygon; in line with the mask used by Winsemius et al. (2018) and the IPCC. While this results in a slightly higher value of the extreme climate outcomes at 2 °C of global warming relative to the multimodel mean at +3 °C (see Extended Data Figure 7), it does not change any main conclusions. Accordingly, we only slightly changed the wording in the following sentence of the discussion: “Across these countries, the **second and third** worst-case outcomes at 2 °C align with the multimodel mean at 2.5 °C, though the worst-case model **reaches** the multimodel mean at 3 °C”.

Best regards,

Emanuele Bevacqua, Erich Fischer, Jana Sillmann, and Jakob Zscheischler

Referee #1 (Remarks to the Author):

This paper considers the best and worst case outcomes at 2degrees of global warming across a range of climate models, highlighting the strong structural model uncertainty in the variables considered. This paper is important for our understanding and communication of possible climate impacts across multiple sectors and provides an important piece of literature. I recommend publishing with minor revisions stated below.

We thank the referee for the very positive feedback.

Model independence and model realism are not considered in the paper: there is a brief discussion at the end but I think this could be important and it is why my following suggestions ask for model names on some of the plots to Figure out which model is behaving in which way. Here is one example of a paper that discussed these important factors: <https://esd.copernicus.org/articles/11/995/2020/>

We fully agree that, in line with the importance of model independence and realism, it is relevant to help readers identify which models behave in which way. Accordingly, we followed the referee's suggestion to make model names easily accessible.

The names of all models, including those corresponding to worst and best cases, were already provided in Extended Data Table 1 (now updated following another referee's suggestion and referenced in the main text when presenting the results of the three main sectors), with the names of worst- and best-case models also shown in Extended Data Figures 1, 3 and 4.

We agree that having model names readily available when inspecting the main figures is helpful. Since it was not feasible to include all model names directly in the main figures of each sector due to graphical and space constraints, we implemented the following changes to improve the accessibility of this information:

- We added names of the best- and worst-case models in the captions of the main figure of each sector (Figures 2, 3 and 4 and Extended Data Figures 7, 8).
- We created a new figure, Extended Data Figure 2 (shown below). In this figure, which was created to also account for the next comment of the referee on SMILEs, we included the names of best- and worst-case models for all sectors, as well as names of SMILEs (whose names and statistics are now provided in a new Extended Data Table 2). This allows readers to directly compare worst- and best-case model names across all sectors in a single image. We referred to this figure in the caption of the main figure of each sector (and, in response to another referee's comment, also in other parts).
- We added a reference to Extended Data Table 1 (containing names of all models) in the caption of the main figure of all five sectors (the table now also has a coloured legend as suggested by the referee, see below).

These changes substantially improve transparency regarding model names.

Implementing the above led to several changes to the caption of the main figure of each sector. For brevity, we report below only the change for the caption of the precipitation sector (Figure 2); the corresponding updates for other sectors are visible in the tracked-changes file.

Fig. 2. Extreme climate outcomes for precipitation extremes in highly populated areas. [...] The models with the highest and lowest 8\% values of f represent the best and worst cases (**from top to bottom: CanESM5, KACE-1-0-G, and IPSL-CM6A-LR (worst) and GFDL-ESM4, MRI-ESM2-0, and INM-CM4-8 (best); see Extended Data Table 1 and Extended Data Figure 2a for model names**). The two vertical lines to the right of the bar show the range of f for two Single Model Initial-condition Large Ensembles (**SMILEs**; coloured for the highest and lowest 8\% values — **see Extended Data Figure 2a for more SMILEs**). [...]

Extended Data Fig. 2. Global climatic impact-driver f from multiple Single Model Initial-condition Large Ensembles (SMILEs) in a 2 °C world. **a**, For precipitation extremes in highly populated areas, the same as the right part of Figure 2b, but with the addition of more SMILEs. Specifically, the left orange bar shows the global climatic impact-driver f used to identify climate outcomes computed for individual climate models in a 2 °C world (each model contributes with one ensemble member; the black thick line shows the multimodel mean). The models with the highest and lowest 8% values of f represent the best and worst cases (model names are provided). The multiple vertical lines to the right of the bar show the range of f for different SMILEs, with each SMILE labeled by its name and the number of ensemble members

in brackets (the range of f is coloured for the highest and lowest 8% values derived from quantiles; note that for some SMILEs with only ten ensemble members, the 8% quantiles may nearly overlap with the minimum or maximum values). **b**, The same as panel a, but for droughts across breadbaskets worldwide. **c**, The same as panel a, but for fire weather extremes over forests; furthermore, highest and lowest values were defined via a 12% instead of an 8% threshold. **d**, The same as panel a, but for heat extremes in low-income countries. **e**, The same as panel a, but for marine heatwaves over fisheries. The f values of all SMILEs are provided in Extended Data Table 2.

(1) The two large ensembles have quite different ranges of internal variability. Many more ensembles exist. Why did you choose this specific two? Would it be worth adding another few to see how they compare? This leads back to my first question around model performance – are some models better than others? (2) Additionally it is worth adding a few lines to the paper discussing how the worst or best case model could be even worse or better if the member selected happens to be an end member on the internal variability spectrum. For the current list of available ensembles see here: <https://egusphere.copernicus.org/preprints/2024/egusphere-2024-3684/>

We added numbers in this comment to refer to its two parts. In part (1), we clarify that the reviewer refers to the difference between the range across ensemble members of the two Single Model Initial-condition Large Ensembles (SMILEs), rather than to the difference between the mean values of the two SMILEs. Here, the range across the ensemble members within each SMILE represents uncertainty due to internal climate variability. The purpose of our analysis is to gain insights into the contribution of internal variability to overall uncertainty (as inferred from the intra-SMILE range) relative to the contribution from model differences. The conclusion stated in the main text is that for the three main sectors, differences between models constitute a relevant (though not the only) source of uncertainty. The same holds for temperature extremes across populated areas, one of the two sectors shown in the Extended Data.

In the first submission, we used two SMILEs, not because these models are better than others, but because they allowed us to support the broad conclusion outlined above. We opted for a setup that enabled us to use the same SMILEs across all sectors. Specifically, we selected CMIP6 models for which we could find at least ten ensemble members, with data (starting from the preindustrial period, after concatenating historical and SSP5-8.5 runs) from the same members for all five analysed sectors. This led to selecting CanESM5 and MIROC6, each providing 50 members. Originally, we used only two SMILEs as we do not expect that adding more SMILEs would alter our conclusion above. In particular, (1) if the ensemble spread of additional SMILEs is similar to that of the two original SMILEs, the results remain unchanged; (2) if it is smaller, the conclusion that model differences are important for uncertainty would be strengthened; and (3) if it is larger, the findings from the two original SMILEs remain valid, making it still difficult to conclude that model differences are not as important.

Nevertheless, extending the analysis to additional SMILEs has benefits. This extension now allows us to make our conclusions even more robust, as the additional SMILEs exhibit a similar

ensemble spread to that of CanESM5 and MIROC6 (see Extended Data Figure 2, provided above in reply to the previous comment). Note that we identified additional CMIP6 models with at least ten ensemble members, which resulted in different models and, sometimes, different numbers of SMILE members across the different sectors. Although a few more SMILEs with data on the native-grid (we often need daily data) exist—as indicated by the reference of the reviewer, which provides a collection of monthly data (including 3 monthly maximum derived from daily data of temperature and precipitation)—our SMILE ensemble of opportunity allows for demonstrating the robustness of the conclusion on uncertainty.

In the paper, we took action in several places, as detailed below. Note that after careful consideration, we decided to maintain two SMILEs in the main Figure of each sector because there is little space for additional SMILEs in the figure. Thus, we created Extended Data Figure 2. This dedicated new figure also allowed for adding the names for best- and worst-case models, as well as SMILE names, thus also addressing the previous comment of the referee.

The statistics of the SMILEs are synthesised in Extended Data Figure 2 (see above) and in the new Extended Data Table 2, shown here:

Model (ensemble member)	Precipitation extremes in populated areas	Droughts over breadbaskets	Fire weather in forests	Heat in low-income countries	High oceanic temperature over fisheries
CanESM5 (r1–25(i1p1f1), r1–25(i1p2f1))	12.96 (15.07, 10.72)	11.92 (16.82, 8.5)	4.03 (4.7, 3.1)	2.66 (2.77, 2.53)	1.65 (1.74, 1.57)
MIROC6 (r1–50(i1p1f1))	8.5 (9.65, 7.39)	6.22 (12.29, 0.69)	3.37 (3.75, 2.94)	3.61 (3.74, 3.39)	1.63 (1.73, 1.54)
ACCESS-ESM1-5 (r1–30(i1p1f1))	–	15.06 (19.86, 11.07)	–	2.14 (2.28, 2)	–
ACCESS-ESM1-5 (r1–10(i1p1f1))	8.31 (8.8, 6.78)	–	–	–	1.68 (1.72, 1.64)
CanESM5-1 (r1–10(i1p1f1), r1–10(i1p2f1))	–	12.31 (16.27, 9.83)	–	–	–
GISS-E2-1-H (r1–5(i1p1f2), r1–5(i1p3f1))	–	2.27 (5.17, -0.62)	–	–	–
MIROC-ES2L (r1–10(i1p1f2))	–	22.93 (28.36, 16.7)	3.02 (3.31, 2.78)	3.05 (3.19, 2.92)	1.75 (1.84, 1.68)
MPI-ESM1-2-LR (r1–30(i1p1f1))	–	6.46 (10.72, 0.7)	3.17 (3.76, 2.57)	1.69 (1.88, 1.5)	–
MPI-ESM1-2-LR (r1–10(i1p1f1))	7.49 (8.31, 6.89)	–	–	–	1.57 (1.62, 1.5)

Extended Data Table 2. Single Model Initial-condition Large Ensembles (SMILEs) used for the analyses in a 2 °C world and associated *f* values. Some models appear in multiple rows because different ensemble members (uniquely identified by the standard CMIP6 ripf index) were used across sectors. Specifically, for a given sector, the hyphen “-” indicates that the model was either not employed or not analysed under the same ensemble member configuration as in other sectors. The reported statistics are the mean (max, min) *f* values across ensemble members. The statistics of the first two SMILEs (CanESM5 and MIROC6) are plotted to the right of the orange bar in panel b of each sector's main figure, while all SMILEs are plotted to the right of the orange bar in Extended Data Figure 2.

We modified the part on SMILEs in the Methods:

“Only when quantifying the contribution of internal climate variability to the uncertainties in projections, we used Single Model Initial-condition Large Ensembles (SMILEs)[References] (Extended Data Table 2), that is, the numbers reported in the text are unaffected by the

SMILEs. We used CanESM5 and MIROC6 (each with 50 ensemble members **available for all analysed sectors**), which are shown as vertical lines on the left and right, respectively, in Figures 2b, 3b, and 4b and Extended Data Figures 7b, 8b; these and additional SMILEs with fewer ensemble members are shown in Extended Data Figure 2.“

We referred to the new Extended Data Figure 2 in the new captions of the main figure of each sector (not shown below for the sake of brevity) and, as shown below, to both Extended Data Figure 2 and Extended Data Table 2 in multiple parts of the paper.

In the precipitation section: “Our results also **indicate** that model improvement could reduce uncertainties in the global climatic impact-driver, as they are largely driven by structural differences between models rather than by internal climate variability (Figure 2b, the total uncertainty shown by the model range on the left orange bar is large compared to the uncertainty from internal climate variability shown by the range **of each of the two Single Model Initial-condition Large Ensembles—SMILEs**; **see Extended Data Figure 2a and Extended Data Table 2 for more SMILEs**).“

In the sentence above, we now use “indicate” instead of “suggest” because using more SMILEs strengthens the statement.

In the soil moisture section: “This very large uncertainty in the global climatic impact-driver f mainly arises from model differences rather than internal climate variability (Figure 3b, compare the range from models over the left orange bar with the range from two SMILEs; **see Extended Data Figure 2b and Extended Data Table 2 for more SMILEs**). ”

In the fire weather section: “This large difference in f across climate model simulations is largely caused by model differences rather than internal climate variability (Figure 4b; **Extended Data Figure 2c and Extended Data Table 2 for more SMILEs**), which indicates the potential for model improvements and uncertainty reduction.”

On part (2) of the comment, we agree. Considering multiple ensemble members from SMILEs indeed allows identifying an even more extreme worst-case climate outcome than those found in the pool of single-member models—for precipitation, heatwaves across low-income countries, and fisheries sectors (Extended Data Figure 2a,d,e). For heatwaves across low-income countries, the best-case outcome among SMILE ensemble members is more favourable than the best case identified from the single-member model pool (Extended Data Figure 2d). This interesting behaviour—which arises because uncertainties stem not only from model differences but also from internal climate variability—becomes more evident thanks to adding new SMILEs.

Inspired by this comment, we also believe that it is worth stressing more explicitly that even additional single-member models could produce more extreme worst-case outcomes at 2 °C. Note: adding a few models to a large multimodel pool has little effect on the multimodel mean, but can easily lead to more outcomes at 2 °C that exceed the multimodel mean at 3°C or 4°C.

Thus, we added some text in the final discussion:

“Accordingly, CMIP models constitute an ensemble of opportunity that may not accurately cover the full range of uncertainty in projections and may not sample the actual worst cases—for example, adding a few independent models could reveal more extreme climate outcomes at +2 °C. Moreover, climate models may have systematic biases, leaving room for a reality that could be more or less severe than any current projection[Reference]. [...] More broadly, while model differences largely contribute to uncertainties, internal climate variability also plays a role. Accordingly, explicitly sampling this variability reveals that climate outcomes could turn out more extreme than those captured by single-member models—both for worst cases (Extended Data Figure 2a,d,e) and best cases (Extended Data Figure 2d).”

Finally, note that alongside adding new SMILEs, we could also add two more models with a single ensemble member for soil moisture; these model additions did not affect any of the conclusions of the paper, though we refined the following sentence in the drought section:

“Ten out of **forty-two** models show climate outcomes at a 2 °C warming that are well beyond the multimodel mean at 4 °C of global warming (Figure 3b).”

Figures in General:

For Figure 2 and associated similar Figs.

Which large ensemble is which on the plots?

We clarified this in the Methods, where we stated:

“We used CanESM5 and MIROC6 (each with 50 ensemble members **available for all analysed sectors**), which are shown as vertical lines on the left and right, respectively, in Figures 2b, 3b, and 4b and Extended Data Figures 7b, 8b; these and additional SMILEs with fewer ensemble members are shown in Extended Data Figure 2.”

As stated in the previous reply, we now also provide the names of all SMILEs in the newly created Extended Data Figure 2. Furthermore, following the suggestion in the comment below on “Ext Table 1”, we created Extended Data Table 2, with the statistics of all SMILEs.

Have you considered putting the names of the end member models on panel b) so the reader knows which model is which?

Please see the reply to the second comment of the referee, where we addressed the same matter.

I find it confusing that c and d only show subsets of a – can you put dots or lines or find some way to show which regions are in both a c and d?

We understand the comment and have considered a possible alternative; however, implementing the suggestion poses graphical challenges for some sectors. For example, it might be possible to highlight breadbaskets in panel a for the drought sector, but it is challenging to do the same for sparsely populated areas (points or contours would mask the values in panel a). Therefore, we opted for keeping the figure as it is. Note that this choice also aligns with the target of panel a and the way it is presented at the beginning of the section of each sector. Specifically, panel a reflects the status quo of the current assessments that focus on the whole globe, and accordingly, panel a is presented without direct reference to the critical areas, while panels c and d focus on the critical sectoral area through our novel approach.

Figure captions: I would find it easier if the important parts were in each caption rather than 'as in Fig 2' otherwise the reader has to keep flipping back and forth for the important details

True — this also aligns with editorial indications. We have revised all captions where this comment applies, that is, two captions in the main text and four captions in the Extended Data.

Line 145 – does it make sense to use this fire forest index everywhere? It seems odd there is no data in Australia where we know there are fires?

The index, fire weather index (FWI) extremes across forested areas, specifically targets ecological damage to forested ecosystems. Therefore, if fires occur primarily in non-forested regions, they fall indeed outside the target of this index. This also aligns with the concept of the recent work of Abatzoglou et al. (2025, Nat. Commun.), which focused on FWI in forests.

While there are indeed frequent fires in Australia, many occur in non-forest landscapes such as grasslands or shrublands. Still, the fires the reviewer refers to may occur in areas that are forested or partially forested.

In practice, classifying a pixel as forest is somewhat criterion-dependent. As a criterion, we currently use: “Forest grid cells were defined where the sum of tree densities (Broadleaf and Needleleaf, both Evergreen and Deciduous) exceeds **the sum of** the density of all other classes for all years during 2010-2019” (note, we added “the sum of” relative to the previous version to increase clarity). According to our criterion, we do have a very limited forested area in Australia.

By testing a less restrictive approach, that is: *Forest grid cells were defined where the sum of tree densities (Broadleaf and Needleleaf, both Evergreen and Deciduous) exceeds 20% of the the sum of the density of all classes (excluding water)—meaning that forests exceeds 20% of the land areas—for all years during 2010-2019*, the forested area expands widely, and slightly in Australia. The resulting limited forest cover in Australia aligns with maps available online. This test also allows us to confirm that, also when using this different criterion, the worst-case models at 2 °C of global warming exceed the multimodel mean at 3 °C (Figure R1). We added to the Methods information about this additional test for robustness, by stating:

“Note that the worst-case models at 2 °C exceed the multimodel mean at 3 °C even with a less restrictive forest definition—namely, where the sum of tree densities exceeds 20% of the sum of the density of all classes (excluding water) for all years during 2010-2019.”

Fig. R1. The same as Figure 3, but based on the criterion described in the text to define forested areas. (Minor technical detail: for better visualising the data, we updated the palette for panels c and d in this and the relative figure in the updated paper to range from -8 to 8, instead of from -6 to 6.)

SSP370 is quite a different pathway to SSP585 can you please redo your comparison in Ext Fig 10 with SSP370

We appreciate this comment. As described in the methods, we used SSP5-8.5 as it allows for inspecting warming levels from moderate to extreme with the same consistent dataset. To test the sensitivity of the results in Extended Data Figure 11, we used SSP2-4.5. Although SSP3-7.0 produces warming rates between SSP2-4.5 and SSP5-8.5, it follows a distinct aerosol pathway (<https://www.nature.com/articles/s41558-023-01883-2>); therefore, we agree that it can be interesting to check Extended Data Figure 11 based on SSP3-7.0.

Although a direct comparison of the two versions of Extended Data Figure 11 across scenarios is somewhat constrained by the smaller number of models available for SSP3-7.0 relative to SSP2-4.5, the following figures (Extended Data Figure 11 and the new Figure R2) show that results appear broadly similar, with only minor differences. Note that some differences are expected due to internal climate variability (especially for the fisheries sector, for which internal variability controls uncertainty in projections), and that slightly different x-axis ranges are used in the figures under the two different SSPs.

Extended Data Fig. 11. Sensitivity of extreme climate outcomes' identification in a 2 $^{\circ}C$ world to using different SSP scenarios. The same as Extended Data Figure 11 (comparing SSP5-8.5 with SSP2-4.5). The caption of the figure follows: **a**, For precipitation extremes in highly populated areas, scatterplot of the global climatic impact-driver f from different models obtained via (x-axis) the SSP5-8.5 scenario from 2015 onward (i.e., as in Figure 2b, left orange

bar) and (y-axis) the SSP2-4.5 scenario from 2015 onward. (Note that different ensemble members of the same model may be used in the two scenarios, which does not compromise comparability since differences due to internal climate variability are present between SSP scenarios even when using ensemble members with the same ripf index.) Grey points along the x-axis and y-axis indicate the values of f for models that are available only for the SSP5-8.5 or SSP2-4.5 scenario, respectively. **b**, The same as panel a, but for droughts across breadbaskets worldwide. **c**, The same as panel a, but for fire weather index extremes across forests. **d**, The same as panel a, but for heat extremes in low-income countries. **e**, The same as panel a, but for marine heatwaves over fisheries; **for this sector, note that differences in f values between the two scenarios are small compared to the overall, average f values. In general, some non-systematic differences between SSP scenarios are expected due to internal climate variability.**

Figure R2. Sensitivity of extreme climate outcomes' identification in a 2 °C world to using different SSP scenarios. The same as Extended Data Figure 11, but comparing SSP5-8.5 with SSP3-7.0.

Code availability – this is not sufficient to reproduce the work done in the paper

We agree. As planned with the editorial board earlier, we have now made the code publicly available on Zenodo and provided the link in the code availability section.

(1) Ext Table 1: I recommend putting in brackets the min/mean or medium/max values across the 2 large ensembles. (2) For CanESM5 anecdotally the p1 and p2 ensembles are quite different. Please check that if you split into these two halves that the statistics are the same. (3) One more suggestion is to add a colorbar to the table and highlight models in ranks blue could be 1: red could be the last one. This is often done for evaluation and would show if some models are always to one side of the distribution. An example of what I mean is here: https://pcmdi.llnl.gov/pmp-preliminary-results/interactive_plot/portrait_plot/enso_metric/enso_metrics_interactive_portrait_plots_v20231121.html

We added numbers in this comment to refer to its two parts. About (1), following the suggestion, we created Extended Data Table 2, where we added the statistics of all SMILEs.

About (2), we also split the CanESM5 ensemble into its p1 and p2 subsets and found similar results, as shown in the new Figure R3 below. Specifically, across all sectors, the p1 and p2 subsets exhibit very similar distributions, with similar medians (dashed lines) and minimum and maximum values (see values in brackets in the legend). Furthermore, the Kolmogorov–Smirnov (KS) tests yield p-values consistently above 0.05 (see legend), indicating no statistically significant differences between the distributions of the two subsets. These results confirm that treating CanESM5 as a single ensemble does not affect our conclusions.

Finally, (3) we modified Extended Data Table 1 as suggested by including colours (see below).

Fig. R3. Distribution of the f values for the five analysed sectors based on the CanESM5 model. Sectors are indicated in the panel titles. The light-red and light-blue shaded areas indicate the p1 and p2 ensembles of the model, respectively. Solid vertical lines indicate the

medians, while the values in brackets in the legend denote the minimum and maximum of each subset. The reported Kolmogorov–Smirnov (KS) p-values are reported in the legend.

Model (ensemble member)	Precipitation extremes in populated areas	Droughts over breadbaskets	Fire weather in forests	Heat in low-income countries	High oceanic temperature over fisheries
1. ACCESS-CM2 (r1i1p1f1)	12.28	9.76	–	2.38	1.72
2. ACCESS-ESM1-5 (r1i1p1f1)	8.71	18.8	–	2.22	1.69
3. AWI-CM-1-1-MR (r1i1p1f1)	–	–	–	1.81	1.52
4. BCC-CSM2-MR (r1i1p1f1)	8.45	37.31	–	2.38	1.6
5. CAMS-CSM1-0 (r1i1p1f1)	–	–	–	–	1.59
6. CAMS-CSM1-0 (r2i1p1f1)	12.58	2.3	–	2.1	–
7. CanESM5 (r1i1p1f1)	14.76	10.97	3.97	2.73	1.66
8. CanESM5-1 (r1i1p1f1)	–	12.41	–	–	–
9. CanESM5-CanOE (r1i1p2f1)	–	14.03	–	–	1.62
10. CAS-ESM2-0 (r1i1p1f1)	–	8.37	–	–	1.69
11. CESM2 (r1i1p1f1)	7.01	8.23	–	–	1.63
12. CESM2-WACCM (r1i1p1f1)	7.48	5.39	–	–	1.64
13. CIESM (r1i1p1f1)	–	6.7	–	–	1.54
14. CMCC-CM2-SR5 (r1i1p1f1)	6.54	13.68	2.35	2.18	1.73
15. CMCC-ESM2 (r1i1p1f1)	6.59	10.82	2.26	2.28	1.77
16. CNRM-CM6-1 (r1i1p1f2)	7.17	3.36	–	2.63	1.6
17. CNRM-CM6-1-HR (r1i1p1f2)	5.78	1.4	–	2.29	1.6
18. CNRM-ESM2-1 (r1i1p1f2)	9.03	51.36	–	2.97	1.58
19. E3SM-1-1 (r1i1p1f1)	–	8.42	–	–	1.78
20. EC-Earth3 (r1i1p1f1)	9.13	21.07	1.78	2.09	1.75
21. EC-Earth3-CC (r1i1p1f1)	10.98	16.39	–	2.11	1.63
22. EC-Earth3-Veg (r1i1p1f1)	9.61	18.12	–	1.93	1.56
23. EC-Earth3-Veg-LR (r1i1p1f1)	9.62	21.3	–	2.15	1.68
24. FGOALS-F3-L (r1i1p1f1)	–	1.23	–	–	1.55
25. FGOALS-g3 (r1i1p1f1)	6.55	3.34	2.44	2.21	1.63
26. FIO-ESM-2-0 (r1i1p1f1)	–	–	–	–	1.5
27. GFDL-CM4 (r1i1p1f1)	7.94	35.54	3.41	2.31	1.66
28. GFDL-ESM4 (r1i1p1f1)	4.23	5.28	6.03	2.65	1.68
29. GISS-E2-1-G (r1i1p1f2)	–	7.78	–	2.02	1.7
30. GISS-E2-1-H (r1i1p1f2)	–	3.21	–	–	–
31. HadGEM3-GC31-LL (r1i1p1f3)	9.71	10.47	5.48	3.12	1.63
32. HadGEM3-GC31-MM (r1i1p1f3)	10.21	14.76	6.51	3.16	1.7
33. IITM-ESM (r1i1p1f1)	7.95	–	–	–	–
34. INM-CM4-8 (r1i1p1f1)	3.96	1.74	2.13	2.39	1.66
35. INM-CM5-0 (r1i1p1f1)	6.5	6.24	1.79	2.18	1.68
36. IPSL-CM6A-LR (r1i1p1f1)	13.14	1.86	2.39	2.53	1.78
37. KACE-1-0-G (r1i1p1f1)	13.65	–	6.2	2.37	1.63
38. KIOST-ESM (r1i1p1f1)	13.11	–	1.52	2.5	1.58
39. MCM-UA-1-0 (r1i1p1f2)	–	0.96	–	–	1.59
40. MIROC-ES2L (r10i1p1f2)	–	25.6	2.97	3.01	1.69
41. MIROC-ES2L (r1i1p1f2)	6.95	–	–	–	–
42. MIROC6 (r1i1p1f1)	7.91	7.84	3.39	3.6	1.63
43. MPI-ESM1-2-HR (r1i1p1f1)	6.6	2.47	3.11	1.73	1.55
44. MPI-ESM1-2-LR (r1i1p1f1)	6.89	4.41	3.4	1.74	1.55
45. MRI-ESM2-0 (r1i1p1f1)	4.04	28.56	1.62	2.6	1.57
46. NESM3 (r1i1p1f1)	8.02	–	–	1.88	1.61
47. NorESM2-LM (r1i1p1f1)	7.49	8.61	–	–	1.71
48. NorESM2-MM (r1i1p1f1)	8.88	7.89	4.31	2.88	1.67
49. TaiESM1 (r1i1p1f1)	7.66	-1.02	5.7	2.59	1.68
50. UKESM1-0-LL (r1i1p1f2)	–	2.43	6.13	–	1.61
Total number of employed models	36	42	22	34	45

Extended Data Table 1. Models used for the analyses in a 2 °C world **and associated f values**. For a given sector, the hyphen "-" indicates that the model (ensemble member, **uniquely identified by the standard CMIP6 ripf index**) was not employed. The values are the global

climatic impact-driver f in panel b of each sector's main figure, with the highest and lowest f values—corresponding to the worst- and best-case models—**underlined and in bold**. For each sector, cell color indicates the model's rank based on deciles derived from available models (see legend). Note that for CAMS-CSM1-0 and MIROC-ES2L, different ensemble members were used across sectors. Extended Data Figure 9 shows a scatterplot of f values across sectors.

Ext Figs 1-5: Is there a better projection you can use as much of the map is currently white space?

About these Extended Data Figures, to maintain consistency across figures of different sectors (as well as across main and Extended Data Figures of a given sector), we prefer to keep the current projection.

Ext Fig 4-5 is the MMM also spatially incoherent in these plots? I was a little confused about what was exactly being plotted. Please check the captions.

According to the updated numbering of the figures, the reviewer refers to Extended Data Figures 5 and 6. Thanks for this comment, we agree that the caption must be improved to clarify the aspects mentioned by the referee. We now use the following captions for the two figures:

“Extended Data Fig. 5. Deviation of spatially incoherent worst-case climate outcomes in a 2 °C world from multimodel mean at 3 °C and 4 °C. a, For precipitation extremes in highly populated areas, the difference between (i) the local worst-case climate outcome in a 2 °C world (that is, at each individual location, the mean of $\Delta Rx5day$ (%) across the 8% worst-case local $\Delta Rx5day$ (%)) and (ii) the multimodel mean (MMM) in a 3 °C world. **Note that for (i), deriving a global map this way yields a spatially incoherent climate outcome, as it ignores spatial dependencies by unrealistically assuming that different locations experience an outcome from a different set of models, whereas the multimodel mean in (ii) follows standard practice by using the same, complete set of available models at all locations. ...In all maps, consistent with the other analyses, both the local worst- and best-case climate outcomes are defined using three models.**”

“Extended Data Fig. 6. Spatially incoherent extreme climate outcomes in a 2 °C world **relative to multimodel mean at 2 °C**. a, For precipitation extremes in highly populated areas, the same as Figure 2c, but the **global worst-case climate outcome in a 2 °C world is spatially incoherent. Specifically, it shows the difference between (i) the spatially incoherent global worst-case climate outcome in a 2 °C world built at each location independently as the mean of $\Delta Rx5day$ (%) across the 8% worst-case local $\Delta Rx5day$ (%) and (ii) the multimodel mean (MMM) in a 2 °C world. Note, the climate outcome in (i) is spatially incoherent as it is built ignoring spatial dependencies in $\Delta Rx5day$ (%) by unrealistically assuming that different locations experience an outcome from a different set of models, whereas the multimodel mean in (ii) follows standard practice by using the same, complete set of available models at all locations.** b, As panel a, but averaging at each location the 8% best

ΔR_{x5day} (%). c, Left orange bar: the range of f values across all models, which is identical to the left orange bar in Figure 2b (this uncertainty range considers spatial dependencies in ΔR_{x5day} (%)). Right green bar: the difference between the f values of two synthetic, **unrealistic** models that at each location take the worst- and best-case ΔR_{x5day} (%) from all considered climate models (**that is, unrealistic global worst- and best-case climate outcomes**), respectively. ... **In all maps, consistent with the other analyses, both the local worst- and best-case climate outcomes are defined using three models.**"

Ext Fig 8: Perhaps replace dots with model numbers? And number the models in Ext Table 1? This would help see model coherency on the plot.

That is a useful addition, and we have implemented it. The new figure (Extended Data Figure 9) is shown below (note that the caption has been largely rewritten to improve clarity, as well as the new information on the numbers). Although some numbers in grey are not distinguishable from each other, the same applied earlier when we employed dots; therefore, we believe this is still a useful addition to the figure.

Extended Data Fig. 9. Correlations across sector-specific global climatic impact-drivers in a 2 $^{\circ}C$ world. Scatterplots showing the relationship between **global climatic impact-driver f values across the five considered sectors, with each panel comparing two sectors. Each model is identified by a number (following the numbering in the first column in Extended Data Table 1). Grey numbers along the x- or y-axis indicate f values for models that are available only for one of the two sectors. No significant Pearson correlations across models are found at the 5% level for any sector pairs. Red lines and the associated top-right corner mark the 92nd (88th for wildfires) percentile of f , calculated using all models available for the specific sector of interest (these percentiles are used to identify extreme climatic outcomes in the study). In contrast, grey lines show the percentiles calculated using only the subset of models simultaneously available for both sectors considered in each panel.**

Minor details:

Line 47 please reword 'besides others' and be more specific

Thanks for spotting this. We removed 'besides others' as the following text indeed already makes clear the key aspect.

Line 75 replace 'other' with 'another'

Thanks, we use "Two additional sectors" now.

Line 78 – please define Pluvial

We improved the text as suggested:

"Pluvial floods in urban areas—**surface-water flooding from intense rainfall that exceeds natural or engineered drainage capacity**—are responsible for a substantial share of the **total** impacts from coastal and inland flooding, ranking among the most impactful natural hazards[Reference]. **Because** pluvial floods are typically triggered by precipitation extremes accumulating over one or a few days, **such** extremes **are** a key proxy for flooding in climate studies[Reference]."

Line 172-186 – I found these sentences hard to follow can you read and revise please

We suspect that the reviewer refers to Lines 172-176 (count as in the original manuscript). We agree that this part should be improved and have modified the text as shown below:

Lines 172-176 (count as in the original manuscript): "Accordingly, **the unrealistic assumption that** every location may experience the worst climate outcomes simulated locally by different models strongly overestimates the global-scale worst case. Similarly, this approach would result in a too optimistic **global-scale** best case. Specifically, constructing **unrealistic** global worst- and best-case climate outcomes **at 2 °C of warming in this way** (Extended Data Figure 6; **see Methods**) inflates uncertainty in the considered global projected climatic impact-driver ***f***—**that is, the difference between *f* values computed on such unrealistic global worst- and best-case climate outcomes**—by about 80 to 290% across the three sectors (Extended Data Figure 6c,f,i). Nevertheless, we find that the actual uncertainty **in *f*, properly estimated based on globally-averaged projections, is still large**—about the same as or even 12 times larger than, depending on the sector, the difference between the multimodel mean at 4 and 2 °C of global warming (Figures 2e, 3e, 4e)."

The following lines 176-186 (count as in the original manuscript), which we report below, also include the following paragraph and appear to us clear, which is why we believe that the reviewer meant to refer to Lines 172-176, rather than Lines 172-186. However, please let us know if we have misinterpreted and clarification is needed in the following lines as well.

Lines 176-186 (count as in the original manuscript): “Consequently, considering model uncertainty at 2 °C warming reveals extreme global climate outcomes across critical areas that exceed the multimodel mean at 3 or 4 °C of global warming (**Figures 2, 3, 4**)—levels of warming that are well beyond the Paris Agreement limits.

Our approach can easily be applied to other globally exposed sectors, for instance, hydro-power generation shortfalls due to low water availability, drought-induced tree mortality, and coastal impacts from sea level rise. Our results suggest that the more complex the climatic impact-driver affecting a sector, the larger the uncertainty and the potential for extreme climate outcomes at a given global warming level. For instance, the drought evolution depends on interacting effects, including changes in atmospheric circulation and clouds that modulate precipitation, surface radiation, and wind—factors that, together with vegetation response to elevated CO₂, alter evapotranspiration[Reference]. In contrast, sectors heavily dependent solely on temperature tend to exhibit lower uncertainty, as temperature-related variables have less room to vary across models at a fixed global warming level. “

Line 232 – please refer to the Table so we know where to find the model list – same in Data Availability

We added this information at line 232.

In the methods I couldn't find how the warming levels were calculated

This information is provided at the end of the Data section of the original manuscript. To clarify, given that this information is methodological rather than related to Data, we have now put this information under the new, dedicated heading “Global warming levels” and further expanded the sentence:

“For each model ensemble member and target warming level, we selected the earliest 30-year window whose area-weighted global-mean temperature exceeds the area-weighted preindustrial (1851–1900) global-mean temperature by at least that target warming level.”

Referee #2 (Remarks to the Author):

SUMMARY OF KEY RESULTS, ORIGINALITY AND SIGNIFICANCE

This is a potentially agenda-setting paper which provides an original take on a 'safe landing' approach to climate risk management. Instead of focusing on a 'cascade of uncertainty' when projecting multi model means and ranges for global climate risks, or indeed on the constraints which would avoid certain undesirable outcomes for global climate change impacts (such as the collapse of ocean currents) the authors focus on an analysis of the potential for extreme outcomes in areas of global importance for the survival of humans and ecosystems respectively. Specifically, some key drivers of climate change impacts that are produced by GCMs in the CMIP6 ensemble dataset are selected and worst case outcomes for each climate model, over these specific areas (which are not necessarily spatially contiguous). By showing that we cannot rule out adverse outcomes for low levels of warming such as 2C, they provide vital new evidence relevant to decision making. The policy relevance of this cannot be underestimated, especially because the authors focus on three local outcomes that are globally relevant, by aggregating local outcomes for (a) heavy 5 day precipitation events in areas where 90% of the human population lives (b) soil moisture deficit in grid cells that form the global breadbasket and (c) enhancement of fire weather in ecologically critical forests.

The findings are alarming: they demonstrate that across multiple sectors, the climate outcomes projected for a 2C world, include a significant number of worse-case scenarios (around 10%) in which the outcomes EXCEED SIGNIFICANTLY the outcomes associated with the multi-model mean outcomes for higher levels of warming. This suggests that a precautionary approach is needed to avoid these outcomes with greater than 90% probability, of limiting level of warming to well below 2C as stipulated in the Paris Agreement. Specifically, the 8% worse case projections of heavy 5 day precipitation in areas where 90% of people live exceeds the multi-model mean for 3C warming; and the four worst case models quantifying exposure of forests to fire weather at 2C warming exceed the multi-model projections at 3C warming. Uncertainty in model projection of soil moisture deficit is found to be very large with a quarter of models projecting outcomes at 2C which exceed the multi model mean at 4C warming.

The originality and policy relevance of the approach convinces me that this paper is appropriate for publication in Nature. The implication of the paper is a far greater need for climate change mitigation in order to avoid potentially serious outcomes for humans and ecosystems with high confidence.

The paper is potentially an agenda setting research paper, since other climate change impact models (with for example models of flood and drought risk modelled hydrologically, models of biodiversity loss and so on) could utilize this safe landing approach to the subsets of the globe where adverse impacts would be particularly important to avoid.

We thank the referee for sharing their detailed perspective on the work and the very positive feedback.

DATA/METHODS: VALIDITY, QUALITY OF DATA, QUALITY OF PRESENTATION, STATISTICS AND TREATMENT OF UNCERTAINTIES

The methods and results are very clearly presented. The data sets selected are widely accepted, publicly available on well known reliable websites of established institutions and organisations that curate and archive climate model output and population data. The quality of the data utilised is state of the art.

We are glad to hear this positive feedback.

*The analysis is sound, although I suggest some improvements/clarifications as follows:
(1) Line 66 states that 10% is used at the cutoff for extreme values of the hazard indicator f . However it is later explained that an 8% cutoff is used for heavy precipitation events and 12% for fire weather. Explain/correct inconsistency.*

The reviewer refers to line 66, where we originally wrote: “We define ‘extreme climate outcomes’ as simulations with the upper and lower ~10% values of f .” The tilde (~) is used as the exact cutoff slightly differs across sectors and at this stage of the manuscript, we originally wanted to avoid technical details (which are provided later on in the manuscript). However, we agree that this could be misleading.

As stated in the Methods, “models with f values below and above the 8th and 92nd percentiles (for wildfires 12th and 88th percentiles given that fewer models are available for FWI) are selected as best and worst-case climate outcomes“. That is, we use an 8% cutoff for all sectors, except 12% for wildfires, as fewer models are available for FWI. While the exact choice of the threshold is subjective, this choice results in three worst and three best cases for all sectors.

To clarify, we modified the original sentence: “We define ‘extreme climate outcomes’ as simulations with the upper and lower ~10% values of f (**8% or 12%, depending on the model availability for the sector; see Methods**).”

(2) In the methods section, a number of acronyms are used to describe ensemble members - these acronyms need to be detailed. Later in extended data table 1, 'riilp1f1' and other character strings require explanation.

We thank the referee for bringing this to our attention. We clarified in the Methods by adding:

“the ripf is an index used in CMIP6 to uniquely identify ensemble members of a given model, where r, i, p, and f denote the realization, initialization method, physics parameterization, and forcing index, respectively”.

Furthermore, in the Extended Data Table 1, we added the text in bold:

“For a given sector, the hyphen "--" indicates that the model (ensemble member, **uniquely identified by the standard CMIP6 ripf index**) was not employed.”

Similarly, in the newly added Extended Data Table 2:

“... ensemble members (**uniquely identified by the standard CMIP6 ripf index**) ...”

Furthermore, after carefully reviewing the entire Methods, we believe that the reviewer may also refer to the technical terms used by CMIP6 to uniquely identify the time resolution and name of the variable. Therefore, we modified the text as:

“We derived time series of annual maximum 5-day precipitation (from daily data; **CMIP6 identifier of the used variable: day, pr**), annual mean soil moisture over the total column (from monthly data; *Lmon, mrso*) [...] computed based on daily precipitation, wind (**day, sfcWind**), relative humidity (**day, hurs**), and maximum temperature. “

(3) Why are the first ensemble members chosen for the analysis? Is this simply meant to be a 'random' selection? Is there any reason why that might not be the case?

The *ripf* convention uniquely distinguishes ensemble members through four indices: *r* = realization, *i* = initialization procedure, *p* = physics parameterization, and *f* = forcing index. In practice, for a given model, the *ipf* indices are very often fixed, while several *r* values may be available (when multiple *ipf* combinations exist, none can generally be considered superior to the others). For a given model, the main choice is therefore among the *r* indices. However, since different *r* values represent simulations that only differ due to internal variability, selecting *r1i1p1f1* instead of, for example, *r2i1p1f1* or *r3i1p1f1*, is effectively random. We used *r1i1p1f1* when available, as it is the only ripf member available for many models and is therefore commonly used in climate studies.

(4) Why is the full ensemble only studied for two GCMs and why those particular two? Are the full ensembles available for the other GCMs, or only for those two? If the scope of the study meant that only two could be included, then it is important to justify why these two in particular were selected.

Given that the referee provided a comment related to the same matter later, we report this comment below and reply to the two related comments together.

Extended figure 5 is particularly useful in showing what would happen if the approach were (incorrectly) applied to everywhere (in each grid cell) simultaneously without considering spatial coherence. It would be helpful to detail mathematically, the formula used to construct this to show how it differs from the formula used to construct figure 2 which is spatially coherent, to make this very clear to a wider range of readers. This is quite critical to convincing the reader that the approach is mathematically sound (by

illustrating how NOT to do it ... the key advance made by the authors is in fact figuring out an alternative to this 'wrong' way of proceeding, and it is why this has not, to my knowledge, been done before.

According to the updated numbering of the figures, the reviewer refers to Extended Data Figure 6. We share the same view of the referee and also find the idea of the formulas very elegant. We worked on writing formulas in a way that intuitively allows the reader to grasp the issue of the incoherent approach, which is due to the incorrect positioning of the maximum operator when deriving the maximum value of f . We expanded the Methods to include these considerations, which also allowed us to more clearly explain why the incoherent approach leads to an overestimation of the uncertainty in Extended Data figure 6c,f,i. In the Methods, we added a subsection:

Coherent vs incoherent climate outcomes

For each sector, deriving extreme climate outcomes as described above implies that the worst-case outcome corresponds to the model m^w associated with the maximum value of f_m (Eq. 1). This yields a *spatially coherent* field, as it represents the projected climatic impact-driver from the same model at all locations x , that is $\Delta CID_x^{(coh,w)} = \Delta CID_{x,m^w}$, where $m^w = \arg \max_m f_m$. The corresponding f value is:

$$f^{coh,w} = \max_m \left(\frac{\sum_{x \in \text{critical area}} a_x \Delta CID_{x,m}}{\sum_{x \in \text{critical area}} a_x} \right). \quad (2)$$

In contrast, in Extended Data Figure 6, unrealistic, *spatially incoherent* worst cases are shown. Here, the most extreme outcome assumes that each grid cell x experiences the maximum projected climatic impact-driver across models, that is $\Delta CID_x^{(incoh,w)} = \max_m \Delta CID_{x,m}$, resulting in an unrealistic global field where different locations experience projections from different models. The corresponding f value is:

$$f^{incoh,w} = \frac{\sum_{x \in \text{critical area}} a_x (\max_m \Delta CID_{x,m})}{\sum_{x \in \text{critical area}} a_x}. \quad (3)$$

The different placement of the *max* operator in $f^{coh,w}$ and $f^{incoh,w}$ illustrates the core problem of the incoherent approach, as applying the maximization before spatial averaging permits different models to determine different locations, undermining spatial consistency and inflating f values. Likewise, the best-case outcome—obtained by replacing *max* with *min*—also breaks spatial consistency and deflates f values under the incoherent approach. As a result of these inflated and deflated f values, the incoherent approach exaggerates the uncertainty in f , defined as the difference between f values for worst- and best-case climate outcomes (Extended Data Figure 6c,f,i).

Furthermore, we referred to this new part of the methods (by adding “**see Methods**”) in the caption of Extended Data Figure 6 and in the final discussion of the main text.

It would also be interesting to create a comparison, in Figure 5, as follows: what would be the outcome if spatial coherence was accounted for, but the metrics are averaged

over the whole global land surface for each climate model and then the worst 10% are extracted? This would show whether the areas where people live and grow crops are disproportionately exposed to climate hazard as compared with the whole of the terrestrial land.

We carried out the suggested analysis and—for the three main sectors considered in the paper—we averaged changes in climatic impact-drivers (CIDs) across all land areas rather than focusing only on critical areas. For droughts, the worst-case models at +2 °C of global warming remain above the multimodel mean at 4 °C, which appear consistent with structural model differences related to, for example, vegetation responses that may be relevant across locations. In contrast, for precipitation extremes, the worst-case model no longer exceeds the multimodel mean at +3 °C, in line with the referee’s intuition. Similarly, for FWI extremes across forests, only one model remains above the multimodel mean at +3 °C, compared to four models found in the original setting.

In all cases, the metric values are more extreme when obtained by averaging CID changes across critical areas than across all land areas. For example, the worst-case model change in precipitation extremes is about 15% when averaged across critical areas and 12% when averaged globally; for droughts, about 51% versus 42%; and for FWI, about 6.5% versus 5.1%, respectively. This pattern is consistent with the fact that extreme changes in CIDs across critical regions can be dampened by compensating weaker changes elsewhere when averaged across all land areas. That is, not conditioning the CID changes to the critical area may be misleading, as it can hide worst-case changes concentrated in critical regions, leading to underestimating changes in regions that matter most for risk and impact assessments.

On a methodological side, as for droughts we computed the global average by considering land masses where all models have data, this resulted in excluding most of Greenland, Antarctica, and Iceland. For precipitation extremes, to maintain consistency with the paper’s analysis, we continue to exclude very dry locations using the same approach explained in the original Methods.

We added a sentence to the paper:

“Focusing on critical sectoral areas is essential, as averaging climatic impact-driver changes globally can mask extremes concentrated in these critical areas, leading to an underestimation of worst-case climate outcomes. “

Finally, we also updated Figure 3a, as stated in the Methods: **“In Figure 3a, Greenland, Antarctica, and Iceland are excluded because these regions contain many locations for which not all models provide data.”** This change does not affect any numbers in the paper, but only Figure 3a. In the same figure, for graphical purposes only, we now cut the bottom part of the palette as values are bounded to not be smaller than -20%.

Extended figure 8 is difficult to understand. Please elaborate the figure caption.

We agree that the caption of this figure (which is Extended Data Figure 9 according to the updated figure numbering) could be improved. We write:

“Extended Data Fig. 9. Correlations across sector-specific global climatic impact-drivers in a 2 °C world. Scatterplots showing the relationship between **global climatic impact-driver f values across** the five considered sectors, **with each panel comparing two sectors. Each model is identified by a number (following the numbering in the first column in Extended Data Table 1). Grey numbers along the x- or y-axis indicate f values for models-(ensemble members) that are available only for one of the two sectors.** No significant Pearson correlations across models are found at the 5% level for any sector pairs. Red lines and the associated top-right corner mark the 92nd (88th for wildfires) percentile of f , **calculated using all models available for the specific sector of interest (these percentiles are used to identify extreme climatic outcomes in the study).** In contrast, **grey lines show the percentiles calculated using only the subset of models simultaneously available for both sectors considered in each panel.**”

The change also includes correctly stating 5%, instead of 10%, for the level that was used to define significant correlations.

Extended figure 10 is useful, but the text explaining how it is derived (line 235) requires more explanation to document how the time slices or years at which 2C warming is reached are extracted from the two time series for the two SSPs.

The reviewer refers to Extended Data figure 11 (according to the updated figure numbering). We created the subsection “Global warming levels” in the Methods, where we explain this:

“For each model ensemble member **and target warming level, we selected the earliest 30-year window whose area-weighted global-mean temperature exceeds the area-weighted preindustrial (1851–1900) global-mean temperature by at least that target warming level. The same procedure was applied to the main time series used in the analyses—based on SSP5-8.5—and to those based on SSP2-4.5 used in Extended Data Figure 11 to test the sensitivity of the results at 2 °C of global warming to the SSP.**”

CONCLUSIONS: ROBUSTNESS, VALIDITY, RELIABILITY

The conclusions are valid, and the implications of the paper, in terms of the methodological insights and advance it provides, would be expected to be robust to the selection of different sets of GCM ensemble members.

We thank the referee for the positive feedback.

The exact numbers might be affected by the choice of the two particular GCMs ensembles, hence my request above that the authors detail whether there was any particular reason for this choice.

(4) Why is the full ensemble only studied for two GCMs and why those particular two? Are the full ensembles available for the other GCMs, or only for those two? If the scope of the study meant that only two could be included, then it is important to justify why these two in particular were selected.

The second of these two comments was provided by the referee earlier, and we moved it here to address both comments together. We thank the referee for raising this matter. It appears that the general comment stems from a misunderstanding that we addressed by providing the needed clarification both in the response below and in the Methods. We also added more Single Model Initial-condition Large Ensembles (SMILEs) to the analysis to fully address the comment, as detailed below.

On the first part of the comment, we clarify that the “choice of the two particular GCMs ensembles”—that is, the two SMILEs—does not affect the numbers in the paper. The analyses in the manuscript rely on about 50 climate models (see Extended Data Table 1), using one ensemble member per model (to give the same weight to each model, given that the number of available ensemble members is model-dependent, with many models having only one member). The two SMILEs—CanESM5 and MIROC6—were used solely to assess the effect of internal climate variability on the uncertainties. Accordingly, these SMILE-based results appeared only as the two vertical thin bars in Figures 2b, 3b, and 4b, and in Extended Data Figures 7b and 8b. (A SMILE consists of multiple ensemble members generated with identical external forcings but differing initial conditions, hence sampling internal variability.) Thus, none of the numbers reported in the text depend on the choice of the SMILEs.

To avoid any ambiguity, we now explicitly state in the Methods what SMILEs are used for:

“Only when quantifying the contribution of internal climate variability to the uncertainties in projections, we used Single Model Initial-condition Large Ensembles (SMILEs)[References] (Extended Data Table 2), that is, the numbers reported in the text are unaffected by the SMILEs. We used CanESM5 and MIROC6 (each with 50 ensemble members available for all analysed sectors), which are shown as vertical lines on the left and right, respectively, in Figures 2b, 3b, and 4b and Extended Data Figures 7b, 8b; these and additional SMILEs with fewer ensemble members are shown in Extended Data Figure 2.”

Concerning the second part of the comment, in the first submission, we opted for a setup that enabled us to use the same SMILEs across sectors. Specifically, we selected CMIP6 models for which we could find at least ten ensemble members, with data (starting from the preindustrial period, after concatenating historical and SSP5-8.5 runs) from the same ensemble members for all five analysed sectors. This criterion led us to use CanESM5 and MIROC6 (50 members). These two SMILEs were sufficient to support our conclusion on the relevance of model differences for uncertainties in projections for the three main sectors (the same holds for temperature extremes across populated areas, in the Extended Data). Adding more SMILEs is not expected to alter this conclusion: similar ensemble spreads within the new SMILEs would

confirm it; smaller spreads would further strengthen it; and even larger spreads would not allow for undermining the finding from the already considered SMILES.

Nevertheless, as anticipated above, to comprehensively address the referee's comment, we added more SMILES to the analysis—resulting in seven SMILES in total across sectors. The results are shown in the newly added Extended Data Figure 2 (see below), with summary statistics provided in Extended Data Table 2 (available in the updated manuscript). The additional SMILES exhibit an ensemble spread similar to that of the first two SMILES (CanESM5 and MIROC6), thereby reinforcing our original conclusion that, for the three main sectors, model differences constitute an important source of uncertainty.

Extended Data Fig. 2. Global climatic impact-driver f from multiple Single Model Initial-condition Large Ensembles (SMILES) in a 2 °C world. a, For precipitation extremes in

highly populated areas, the same as the right part of Figure 2b, but with the addition of more SMILEs. Specifically, the left orange bar shows the global climatic impact-driver f used to identify climate outcomes computed for individual climate models in a 2 °C world (each model contributes with one ensemble member; the black thick line shows the multimodel mean). The models with the highest and lowest 8% values of f represent the best and worst cases (model names are provided). The multiple vertical lines to the right of the bar show the range of f for different SMILEs, with each SMILE labeled by its name and the number of ensemble members in brackets (the range of f is coloured for the highest and lowest 8% values derived from quantiles; note that for some SMILEs with only ten ensemble members, the 8% quantiles may nearly overlap with the minimum or maximum values). **b**, The same as panel a, but for droughts across breadbaskets worldwide. **c**, The same as panel a, but for fire weather extremes over forests; furthermore, highest and lowest values were defined via a 12% instead of an 8% threshold. **d**, The same as panel a, but for heat extremes in low-income countries. **e**, The same as panel a, but for marine heatwaves over fisheries. The f values of all SMILEs are provided in Extended Data Table 2.

We referred to the new Extended Data Figure 2 in the new captions of the main figure of each sector (not shown below) and, as shown below, we referred to both Extended Data Figure 2 and Extended Data Table 2 in multiple parts of the paper.

In the precipitation section: “(Figure 2b, the total uncertainty shown by the model range on the left orange bar is large compared to the uncertainty from internal climate variability shown by the range of **each of** the two Single Model Initial-condition Large Ensembles—SMILEs; **see Extended Data Figure 2a and Extended Data Table 2 for more SMILEs**).”

In the soil moisture section: “This very large uncertainty in the global climatic impact-driver f mainly arises from model differences rather than internal climate variability (Figure 3b, compare the range from models over the left orange bar with the range from two SMILEs; **see Extended Data Figure 2b and Extended Data Table 2 for more SMILEs**). ”

In the fire weather section: “This large difference in f across climate model simulations is largely caused by model differences rather than internal climate variability (Figure 4b; **Extended Data Figure 2c and Extended Data Table 2 for more SMILEs**), which indicates the potential for model improvements and uncertainty reduction.

SUGGESTED IMPROVEMENTS, FOR POSSIBLE REVISION

Given the size of the ISIMIP community, consider moving Figure 9 into the main text.

This is a valuable suggestion; however, it would result in too many figures for the format of an Article in the physical sciences category, which is why we opted to include the figure in the Extended Data.

The authors should acknowledge more fully the gap between a single measure of heavy precipitation and a flood risk outcome; similarly the gap between a soil moisture deficit

and a crop failure. I suggest that the authors modify the wording in the paper to explain that they are quantifying exposure to climatic hazard, rather than the risk of an adverse outcome itself. The vulnerability of the human or natural systems is affected by physical processes mediating the response to moisture deficit or excess, for example, soil texture or river flows between grids. Beyond this, risk itself is of course mediated by the vulnerability and potential to adapt to the changes. This is of course beyond the scope of the paper, and I am not suggesting that the authors should extend their calculations to cover these aspects, but they should acknowledge them in more detail.

We thank the reviewer for this constructive comment. In line with the text of the paper, we agree that our study does not focus on *risk* in the IPCC sense, which is a function of hazard, exposure, and vulnerability. After careful discussion among coauthors—several of whom contribute to IPCC reports—we adopted the term ‘*global sector-mean projected climatic impact-driver*’ (hereinafter, ‘*global climatic impact-driver*’) to refer to our metric *f*, thereby rigorously anchoring on the standard IPCC usage of *climatic impact-driver (CID)*. This terminology is used to refer specifically to the hazard-related aspects of the risk framework and to emphasise that our focus is on the physical hazard component relevant to a given sector. Accordingly, we refer to the extreme models identified by ranking of the *f* as “extreme climate outcomes” (rather than, for example, “high impact/risk outcomes”), to make explicit the focus on the climate—rather than the impact/risk—domain.

We would also like to clarify that, while we agree with the reviewer’s general point, our approach is not “quantifying exposure to climatic hazard”, but rather the severity of sector-relevant climatic impact-drivers across regions where exposure exists. In other words, our global climatic impact-driver focuses on exposed areas (e.g., key breadbasket regions or populated areas) to assess the hazard component of the risk. We agree that the vulnerability of affected systems and the magnitude of realised impacts depend on additional biophysical and socio-economic processes, which are beyond the scope of this study.

A discussion on the relevance of linking our results to impact modelling is provided in the final section. Text on the differentiation of climatic impact-drivers and impacts/risk is provided in various parts of the original manuscript; please see examples in Table R1 below. Nonetheless, we agree that adding some text to further clarify the aspect raised by the referee can help, both in general and specifically for the precipitation and drought sectors.

Therefore, we added a general statement when introducing the global climatic impact-driver:

“Finally, *f* quantifies sector-relevant climatic impact-drivers across critically exposed areas, whereas impacts and associated risks arise from the complex interaction of climatic impact-drivers with exposure and vulnerability.”

In the section on precipitation extremes:

“To inform on impact-relevant events, here we only consider precipitation extremes across populated areas, as human settlements are a prerequisite for most flooding impacts. **Such exposed populated areas—where vulnerability to flooding depends on factors like drainage capacity—**are also relevant for the reinsurance sector, which typically focuses on exposed areas to assess financial risks[References].

In the section on droughts:

“Yet, regardless of pending plausibility checks, these results also call for examining the potential impacts of extreme climate outcomes on global crop production **through crop modelling, which can incorporate vulnerability factors such as irrigation, soil properties, and complex management practices.**”

Furthermore, we modified the second last paragraph of the discussion in view of the next comment of the referee (see new text in the next reply), which considers impact modelling explicitly.

Abstract	“Our approach can easily be adapted to a wide range of sectors to support the improvement of sector-specific climate risk assessment and to inform climate policy.”
Part on the global climatic impact-driver	Noth in the section “An approach for identifying extreme future global climate outcomes” and in the caption of Figure 1, we state that the global climatic impact-driver f , “serves as a proxy for projected potential impacts on the specific sector” .
Section on precipitation extremes	“Because pluvial floods are typically triggered by precipitation extremes accumulating over one or a few days, such extremes are a key proxy for flooding in climate studies [Reference].”
Section on droughts	“Concurrent droughts across breadbaskets are often used as an indicator to assess future climate risks to global agriculture and food security[References].” “Yet, regardless of pending plausibility checks, these results also call for examining the potential impacts of extreme climate outcomes on global crop production”
Section on fire weather	“While human activities significantly influence fire dynamics [Reference], fire-prone weather conditions are a fundamental driver of wildfires.”
Discussion	When referring to sea surface extremes in fisheries: “fish mortality is likely also influenced by short-term marine heatwaves, deoxygenation, nutrient shortage, pollution, overfishing, and fish migrations [References].” “Pending these checks, however, the potential implications of extreme climate outcomes must be acknowledged transparently and factored into decision-making.” “Our simple approach could be the basis for a more systematic strategy to guide impact modellers in the selection of a subset of climate models that sample the full range of possible climate outcomes”

	The second-last paragraph includes a direct discussion on the integration of our results on extreme climate outcomes with impact/risk modelling (see reply to next comment for text on this matter).
--	--

Table R1. Examples from the original manuscript where text (underlined) refers to the difference between climatic impact-drivers and impacts/risk.

The authors specifically single out the ISIMIP protocol, including its basis on 5 single GCMs, to show the inadequacy of the approach in the light of this analysis. However, the authors also argue that the approach should be applied by risk assessment studies more generally. For example, a flood risk model creating hydrological projections across CMIP6 ensemble could instead focus on these heavily populated regions and explore these extreme outcomes. The authors might go a little further in recommending how their approach might be applied to ‘climate impact models’, the running of which create many of the projects used in IPCC assessment reports, which have tended to focus on global or regional scale analysis. A more sectoral based approach is also appropriate for adaptation studies – these are the worst outcomes which may need to be planned for.

The authors would do well to explain that their approach is important for informing both climate change mitigation policy and also climate change adaptation policy.

We agree that expanding the text on this matter can be helpful, as well as highlighting more of the climate adaptation aspect. We modified the last paragraph of the discussion, which has been split into two paragraphs. We now write:

“Our simple approach could be the basis for a more systematic strategy to guide impact modellers in the selection of a subset of climate models that sample the full range of possible climate outcomes[References] for a given impact sector. Currently, large-scale initiatives such as the latest protocol of the Inter-Sectoral Impact Model Intercomparison Project (ISIMIP) rely on a limited subset of climate models that likely omits the best- and worst-case climate models. Hence, ISIMIP-based simulations probably underestimate the range of possible global climate impacts for many sectors at 2 °C warming (Extended Data Figure 10). **This limitation highlights challenges in large-scale impact-modelling initiatives, possibly stemming from the high computational cost of impact simulations and the need for automated workflows that can handle large ensembles of climate model inputs.** Given the need to carefully assess the consequences of worst-case climate outcomes, a concerted effort among stakeholders, impact modellers, and climate scientists is essential to identify plausible evolutions of extreme global impacts that can support well-informed decision-making[Reference]. **Our approach could also help guide a more targeted use of climate models in impact modelling and risk assessments. For example, rather than relying on uniform subsets of climate models across sectors, flood-risk or agricultural impact modellers could—in collaboration with climate scientists and stakeholders—use our approach to identify sector-specific extreme outcomes that merit explicit simulation and to focus on global critical exposed areas. This would enable exploring currently underrepresented extreme outcomes—those that are very relevant for risk management**

and adaptation planning. While our indicator *f* represents an initial step for identifying extreme climate outcomes, more advanced low-dimensional indicators—potentially derived via explainable machine learning that links climate drivers to impact data—could further improve the identification of worst-case climate outcomes and optimise model selection for impact simulations. At the IPCC level, strengthening interactions between working groups I and II[References], which respectively assess physical climate changes and their impacts, is key for informing climate risk management **and both mitigation and adaptation policy** under these large uncertainties.

We have demonstrated that, across multiple sectors, global climate outcomes at 2 °C warming could be much more extreme than those expected in a 3 °C or 4 °C warmer world. As global warming approaches 1.5 °C[Reference], this study highlights the urgency of ambitious mitigation strategies aimed at limiting global warming well below 2 °C, emphasising that even a 2 °C warmer world does not necessarily safeguard against severe sectoral global climate impacts.”

REFERENCES – APPROPRIATE CREDIT TO PREVIOUS WORK?

An extended reference list provides context. However it would be helpful to cite the literature on climate risk hotspots, eg Byers, E., et al., 2018: Global exposure and vulnerability to multi-sector development and climate change hotspots. Environ. Res. Lett., 13(5), 55012, doi:10.1088/1748-9326/aabf45. It would also be useful to cite some of the literature which projects climate risk that are based largely on multi model means, e.g. Arnell, N.W., J.A. Lowe, B. Lloyd-Hughes and T.J. Osborn, 2018: The impacts avoided with a 1.5°C climate target: a global and regional assessment. Clim. Change, 147(1), 61–76, doi:10.1007/s10584-017-2115-9; Warren, R., Andrews, O., Brown, S., et al. 2022. Quantifying implications of limiting global warming to 1.5 or 2°C above pre-industrial levels. Climatic Change 172: 39

We thank the referee for suggesting these relevant references. Unfortunately, we were already above the reference limit provided in the editor guidelines (50 references), which prevents us from adding new references to the manuscript. In fact, we had to reduce the number of references.

CLARITY AND CONTEXT, LUCIDITY OF ABSTRACT/SUMMARY, APPROPRIATENESS OF ABSTRACT, INTRODUCTION AND CONCLUSIONS

The paper is generally extremely well written, and is in the style appropriate for publication in Nature. The abstract and introduction are incisive and easy to understand for the general reader. The conclusions are appropriate, but could be extended as recommended above.

We thank the referee again for the very positive and constructive feedback.